# MAKE THE PERTINENT SALIENT: TASK-RELEVANT RECONSTRUCTION FOR VISUAL CONTROL WITH DISTRACTIONS

## ABSTRACT

Recent advancements in Model-Based Reinforcement Learning (MBRL) have made it a powerful tool for visual control tasks. Despite improved data efficiency, it remains challenging to train MBRL agents with generalizable perception. Training in the presence of visual distractions is particularly difficult due to the high variation they introduce to representation learning. Building on DREAMER, a popular MBRL method, we propose a simple yet effective auxiliary task to facilitate representation learning in distracting environments. Under the assumption that task-relevant components of image observations are straightforward to identify with prior knowledge in a given task, we use a segmentation mask on image observations to only reconstruct task-relevant components. In doing so, we greatly reduce the complexity of representation learning by removing the need to encode task-irrelevant objects in the latent representation. Our method, Segmentation Dreamer (SD), can be used either with ground-truth masks easily accessible in simulation or by leveraging potentially imperfect segmentation foundation models. The latter is further improved by selectively applying the reconstruction loss to avoid providing misleading learning signals due to mask prediction errors. In modified DeepMind Control suite (DMC) and Meta-World tasks with added visual distractions, SD achieves significantly better sample efficiency and greater final performance than prior work. We find that SD is especially helpful in sparse reward tasks otherwise unsolvable by prior work, enabling the training of visually robust agents without the need for extensive reward engineering.

## 1 INTRODUCTION

Recently, model-based reinforcement learning (MBRL) (Sutton, 1991; Ha & Schmidhuber, 2018; Hafner et al., 2019; 2020; Hansen et al., 2022; 2023) has shown great promise in learning control policies, achieving high sample efficiency. Among recent advances, the DREAMER family (Hafner et al., 2020; 2021; 2023) is considered a seminal work, showing strong performance in diverse visual control environments. This is accomplished by a close cooperation between a world model and an actor–critic agent. The world model learns to emulate the environment's forward dynamics and reward function in a latent state space, and the agent is trained by interacting with this world model in place of the original environment.

Under this framework, accurate reward prediction is all we should sufficiently require for agent training. However, learning representations solely from reward signals is known to be challenging due to their lack of expressiveness and high variance (Hafner et al., 2020; Jaderberg et al., 2017). To address this, DREAMER employs image reconstruction as an auxiliary task in world model training to facilitate representation learning. In environments with little distraction, image reconstruction works effectively by delivering rich feature-learning signals derived from pixels. However, in the presence of distractions, the image reconstruction task pushes the encoder to retain all image information, regardless of its task relevance. For instance, moving backgrounds in observations in Fig. 1 are considered distractions. Including this information in the latent space complicates dynamics modeling and degrades sample efficiency by wasting model capacity and drowning the relevant signal in noise (Fu et al., 2021).

Figure 1: *Left*: Providing mask example(s) and fine-tuning a mask model, or instrumenting a simulator, to obtain masks. *Right*: An input observation in a distracting Meta-World with three alternative auxiliary task targets. Moving scenes in the background are considered distractions. (b) Observations including task-irrelevant information, disturbing world-model training. (c) and (d) Segmentation of task-relevant components using, respectively, a ground-truth mask and an approximate mask generated by segmentation models.

Distractions are prevalent in real-world visual control tasks. A robot operating in a cluttered environment such as a warehouse may perceive much task-irrelevant information that it needs to ignore. When training with domain randomization for added policy robustness, task-irrelevant information is actively added and must be denoised. Prior approaches (Zhang et al., 2021; Nguyen et al., 2021; Deng et al., 2022; Fu et al., 2021; Bharadhwaj et al., 2022) address the noisy reconstruction problem by devising reconstruction-free auxiliary tasks, such as contrastive learning (Chen et al., 2020). However, many of them suffer from sample inefficiency, requiring many trajectories to isolate the task-relevant information that needs to be encoded. Training with such methods becomes particularly challenging in sparse reward environments, where the signal for task relevance is very weak. Additionally, working with small objects, which is common in object manipulation tasks, poses difficulties for these methods because those objects contribute less to loss functions and are easily overlooked without special attention (Seo et al., 2022).

Inspired by these problems, we address the following question in this paper: *How can we help world models learn task-relevant representations more efficiently?* Our proposed solution takes advantage of the observation that identifying task-relevant components within images is often straightforward with some domain knowledge. For instance, in a robotic manipulation task, the objects to manipulate and the robot arm are such task-relevant components, as shown in Fig. 1 (Left). Given this assumption, we introduce a simple yet effective alternative auxiliary task to reconstruct only the task-related components of image observations. We accomplish this by using segmentation masks of task-related objects which are easily accessible in simulations. Specifically, we replace Dreamer's auxiliary task to reconstruct raw RGB image observations with an alternative task to reconstruct images with a *task-relevance mask* applied to them. (Fig. 1c). By doing this, the world model can learn features from a rich pixel-reconstruction loss signal without being hindered by the noise of visual distractions.

In contrast to previous work that incorporates segmentation masks in reinforcement learning (RL) as an input (James et al., 2019; So et al., 2022), we only use masks in an auxiliary task to improve representation learning. This brings about several advantages. First, we only need segmentation masks during training. The inputs for our method are still the original (potentially distracting) images, so masks are not needed at test time, making our method more computationally efficient. Moreover, the masks we use do not need to be perfect. Since they are used as a target for an auxiliary task, these masks can contain errors as long as they guide feature learning to be informative for the downstream task, which leaves room to replace a ground-truth (GT) mask with its approximation.

To this end, we present a way of training with our auxiliary task with segmentation estimates. This can be useful in many practical cases where no GT mask is available during training. This is made possible by recent advances in segmentation foundation models (Kirillov et al., 2023; Zhang et al., 2023; Xie et al., 2021). Specifically, we fine-tune segmentation models with annotated training data and use them to generate pseudo-labels for the auxiliary task. Fig. 1d shows an example of an auxiliary target made from segmentation prediction. Although the performance with segmentation estimates is impressive without further modification, we find that the training can sometimes

be destabilized by incorrect learning signals induced by segmentation prediction errors. Thus, we additionally provide a strategy to make training more robust to prediction errors and achieve higher performance. Specifically, our strategy is to identify pixels where the foundation model mask prediction disagrees with a second mask prediction given by our world model. We ignore the RGB image reconstruction loss on these pixels to avoid training on potentially incorrect targets.

As previously mentioned, our method assumes that task-relevant parts are easy to identify in image observations with prior knowledge. This is not a strong assumption in many object-centric and robotic domains, where image observations can often be decomposed into relevant and irrelevant regions. However, there are scenarios beyond our scope, where this assumption may not hold because prior knowledge is unavailable or difficult to codify, such as Atari (Bellemare et al., 2013).

We evaluate our method on various robotics benchmarks, including DeepMind Control Suite (Tassa et al., 2018) and Meta-World (Yu et al., 2019), perturbing both with visual distractions. We show that our method for reconstructing masked RGB targets using the ground-truth masks in the presence of distractions can reach the same level of performance as training in *original* environment with no distractions added. Our method for training with approximate masks also shows impressive performance, often matching the performance of the ground-truth mask variant. In both benchmarks, our approximate-mask method achieves higher sample efficiency and superior test returns compared to previous approaches. Notably, this is accomplished with very few task-specific mask example data points (1, 5, or 10 used for fine-tuning), with much of its strength coming from the power of segmentation foundation models. Our method effectively addresses the challenge of training agents in distracting environments by offloading the identification of task-relevant regions to out-of-the-box segmentation models, thereby achieving great sample efficiency and generalization ability. Furthermore, our method proves particularly effective in sparse reward environments and those involving small objects, where prior approaches often struggle.

## 2 RELATED WORK

**Model-Based RL for Distracting Visual Environments.** Recent advances in MBRL have facilitated the learning of agents from image observations (Finn & Levine, 2017; Ha & Schmidhuber, 2018; Hafner et al., 2019; 2020; 2021; 2023; Schrittwieser et al., 2020; Hansen et al., 2022; 2023). Nevertheless, learning perceptual representations in the presence of distractions remains challenging in these models (Zhu et al., 2023). For effective representation learning, some works apply non-reconstructive representation learning methods (Nguyen et al., 2021; Deng et al., 2022), such as contrastive learning (Chen et al., 2020) and prototypical representation learning (Caron et al., 2020). However, features learned with these methods do not necessarily involve task-related content since they do not explicitly consider task-relevance in feature learning. Some other works design auxiliary objectives to explicitly use downstream task information (Zhang et al., 2021; Fu et al., 2021). For example, DBC (Zhang et al., 2021) uses a bisimulation metric (Ferns et al., 2011) to encourage two trajectories of similar behaviors to become closer in a latent space. Perhaps most relevant to our method is TIA (Fu et al., 2021) which explicitly separates task-relevant and irrelevant branches to distinguish reward-correlated visual features from distractions. Features from each branch are combined later to reconstruct the original, distracting observation. Recently, a few approaches proposed to leverage inductive biases such as predictability (Zhu et al., 2023) and controllability (Wang et al., 2022; Bharadhwaj et al., 2022) to learn useful features for visual control tasks. These methods have shown to be more effective than using the reward signal alone, but many of them suffer from sample inefficiency, requiring many samples to implicitly identify what is task-relevant from data. In contrast, our work proposes to leverage domain knowledge in the form of image masks to provide an explicit signal for identifying task-relevant information. Notably, training in sparse reward environments with distraction has remained unsolved in the literature. Several methods for robust representation learning have also been proposed for model-free RL (Laskin et al., 2020; Kostrikov et al., 2021; Yarats et al., 2021; Hansen et al., 2021; Hansen & Wang, 2021; Nair et al., 2022; Zhang et al., 2019). However, the results suggest that MBRL is more powerful and sample efficient for visual control tasks, thus we focus on comparison with methods in the MBRL framework.

**Segmentation for RL.** Segmentation models (He et al., 2017; Redmon et al., 2016) have been used in many downstream tasks, including RL, to assist in pre-processing inputs (Kirillov et al., 2023; Anantharaman et al., 2018; Yuan et al., 2018; James et al., 2019; So et al., 2022). Recently,

using segmentation masks in new domains has been made easier by the introduction of one/few-shot masks foundation models (Zhang et al., 2023; Xie et al., 2021) which can quickly adapt to new use-cases. In the context of RL, the prevalent way to use segmentation models is to turn the input modality from RGB images to segmentation masks (James et al., 2019; So et al., 2022). By converting RGB images into semantic masks, agents can effectively handle complicated scenes and thus also be trained with domain randomization. However, processing the input observation requires additional computation at execution time, and an agent trained with predicted segmentation masks can easily break when the segmentation model malfunctions. Our approach, on the other hand, leverages segmentation masks for an auxiliary task, removing the need for the segmentation model after deployment while also achieving higher test-time performance. FOCUS (Ferraro et al., 2023) is another method that uses masked input as an auxiliary target. However, this is primarily devised for disentangled representation, not for handling distractions. Moreover, it only provides preliminary results with segmentation models and lacks results and analysis on the effects on downstream tasks.

Prior works (Wang et al., 2023; Zhong et al., 2024) integrate segmentation models with RL by pre-processing input observations to isolate task-relevant components. While effective, these methods heavily rely on segmentation model quality at test time, making them vulnerable to failures in unfamiliar scenarios that disrupt agent performance. Also, methods such as Zhong et al. (2024); So et al. (2022) require extensive fine-tuning on large synthetic datasets, resulting in substantial training overhead, and introduce high computational costs during inference. By leveraging off-the-shelf segmentation models with as few as 1 to 10 examples, our approach reduces training requirements while maintaining robustness and runtime efficiency during deployment.

## 3 PRELIMINARIES

We consider a partially observable Markov decision process (POMDP) formalized as a tuple $(\mathcal{S}, \Omega, \mathcal{A}, \mathcal{T}, \mathcal{O}, p_0, \mathcal{R}, \gamma)$, consisting of states $s \in \mathcal{S}$, observations $o \in \Omega$, actions $a \in \mathcal{A}$, state transition function $\mathcal{T} : \mathcal{S} \times \mathcal{A} \to \Delta(\mathcal{S})$, observation function $\mathcal{O} : \mathcal{S} \to \Omega$, initial state distribution $p_0$, reward function $\mathcal{R} : \mathcal{S} \times \mathcal{A} \to \mathbb{R}$, and discount factor $\gamma$. At time $t$, the agent does not have access to actual world state $s_t$, but to the observation $o_t = \mathcal{O}(s_t)$, which in this paper we consider to be a high-dimensional image. Our objective is to learn a policy $\pi(a_t | o_{\leq t}, a_{<t})$ that achieves high expected discounted cumulative rewards $\mathbb{E}[\sum_t \gamma^t r_t]$, with $r_t = \mathcal{R}(s_t, a_t)$ and the expectation over the joint stochastic process induced by the environment and the policy.

**DREAMER** (Hafner et al., 2020; 2021; 2023) is a broadly applicable MBRL method in which a world model is learned to represent environment dynamics in a latent state space $(h, z) \in \mathcal{H} \times \mathcal{Z}$, consisting of deterministic and stochastic components respectively, from which rewards, observations, and future latent states can be decoded. The components of the world model are:

$$
\begin{aligned}
&\text{Sequence model:} && h_t = f_\phi(h_{t-1}, z_{t-1}, a_{t-1}) \\
&\text{Observation encoder:} && z_t \sim q_\phi(z_t | h_t, o_t) \\
&\text{Dynamics predictor:} && \hat{z}_t \sim p_\phi(\hat{z}_t | h_t) \\
&\text{Reward predictor:} && \hat{r}_t \sim p_\phi(\hat{r}_t | h_t, z_t) \\
&\text{Continuation predictor:} && \hat{c}_t \sim p_\phi(\hat{c}_t | h_t, z_t) \\
&\text{Observation decoder:} && \hat{o}_t \sim p_\phi(\hat{o}_t | h_t, z_t),
\end{aligned}
\tag{1}
$$

where the encoder maps observations $o_t$ into a latent representation, the dynamics model emulates the transition distribution in latent state space, the reward and continuation models respectively predict rewards and episode termination, and the observation decoder reconstructs the input. The concatenation of $h_t$ and $z_t$, *i.e.* $x_t = [h_t; z_t]$, serves as the model state. Given a starting state, an actor–critic agent is trained inside the world model by rolling out latent-state trajectories. The world model itself is trained by optimizing a weighted combination of three losses:

$$
\mathcal{L}(\phi) \doteq \mathbb{E}_{q_\phi} \left[ \sum_{t=1}^{T} (\beta_{\text{pred}} \mathcal{L}_{\text{pred}}(\phi) + \beta_{\text{dyn}} \mathcal{L}_{\text{dyn}}(\phi) + \beta_{\text{rep}} \mathcal{L}_{\text{rep}}(\phi)) \right]
\tag{2}
$$

$$
\mathcal{L}_{\text{pred}}(\phi) \doteq -\ln p_\phi(o_t | z_t, h_t) - \ln p_\phi(r_t | z_t, h_t) - \ln p_\phi(c_t | z_t, h_t)
\tag{3}
$$

$$
\mathcal{L}_{\text{dyn}}(\phi) \doteq \max(1, \text{KL}[\llbracket q_\phi(z_t | h_t, o_t) \rrbracket \| \quad p_\phi(\hat{z}_t | h_t))])
\tag{4}
$$

$$
\mathcal{L}_{\text{rep}}(\phi) \doteq \max(1, \text{KL}[q_\phi(z_t | h_t, o_t) \quad \| \quad \llbracket p_\phi(\hat{z}_t | h_t) \rrbracket]),
\tag{5}
$$

where $[\![\cdot]\!]$ denotes where gradients are stopped from backpropagating to the expression in brackets.

Critically, the first component of $\mathcal{L}_{\text{pred}}$ for reconstructing observations from world model states is leveraged as a powerful heuristic to shape the features in the latent space. Under the assumption that observations primarily contain task-relevant information, this objective is likely to encourage the latent state to retain information critical for the RL agent. However, the opposite can also be true. If observations are dominated by task-irrelevant information, the latent dynamics may become more complex by incorporating features impertinent to decision-making. This can lead to wasted capacity in the latent state representation (Lambert et al., 2020), drown the supervision signal in noise, and reduce the sample efficiency.

**Problem Setup.** We consider environments where the latter case is true and observations contain a large number of spurious variations (Zhu et al., 2023). Concretely, we consider some features of states $s_t \in \mathcal{S}$ to be irrelevant for the control task. We assume that states $s_t$ can be decomposed into task-relevant components $s_t^+ \in \mathcal{S}^+$ and task-irrelevant components $s_t^- \in \mathcal{S}^-$ such that $s_t = (s_t^+, s_t^-) \in \mathcal{S} = \mathcal{S}^+ \times \mathcal{S}^-$. We follow prior work in visual control under distraction and assume that (1) the reward is a function only of the task-relevant component, *i.e.* $\mathcal{R} : \mathcal{S}^+ \times \mathcal{A} \to \mathbb{R}$; and (2) the forward dynamics of the task-relevant part only depends on itself, $s_{t+1}^+ \sim \mathcal{T}(s_{t+1}^+|s_t^+, a_t)$ (Zhu et al., 2023; Fu et al., 2021; Bharadhwaj et al., 2022). Note that observations $o_t$ are a function of both $s_t^+$ and $s_t^-$, thus we have $\mathcal{O} : \mathcal{S}^+ \times \mathcal{S}^- \to \Omega$.

Our goal is to learn effective latent representations $[h_t; z_t]$ for task control. Ideally, this would mean that the world model will only encode and simulate task-relevant state components $s_t^+$ in its latent space without modeling unnecessary information in $s_t^-$. To learn features pertaining to $s_t^+$, image reconstruction can provide a rich and direct learning signal, but only when observation information about $s_t^+$ is not drowned out by other information from $s_t^-$. To overcome this pitfall, we propose to apply a heuristic filter to reconstruction targets $o_t$ with the criteria that it minimizes irrelevant information pertaining to $s_t^-$ while keeping task-relevant information about $s_t^+$.

# 4 METHOD

We build on DREAMER-V3 (Hafner et al., 2023) to explicitly model $s_t^+$ while attempting to avoid encoding information about $s_t^-$. In Section 4.1, we describe how we accomplish this by using domain knowledge to apply a task-relevance mask to observation reconstruction targets. In Section 4.2 we describe how we leverage segmentation mask foundation models to provide approximate masks over task-relevant observation components. Finally, in Section 4.3, we propose a modified decoder architecture and objective to mitigate noisy learning signals from incorrect mask predictions.

## 4.1 USING SEGMENTATION MASKS TO FILTER IMAGE TARGETS

We first introduce our main assumption, that the task-relevant components of image observations are easily identifiable with domain knowledge. In many real scenarios, it is often straightforward for a practitioner to know what the task-related parts of an image are, *e.g.* objects necessary for achieving a goal in object manipulation tasks. With this assumption, we propose a new reconstruction-based auxiliary task that leverages domain knowledge of task-relevant regions. Instead of reconstructing the raw image observations (Fig. 1b) which may contain task-irrelevant distractions, we apply a heuristic task-relevance segmentation mask over the image observation (Fig. 1c) to exclusively reconstruct components of the image that are pertinent to control.

Since our new masked reconstruction target should contain only image regions that are relevant for achieving the downstream task, our world model should learn latent representations where a larger portion of the features are useful to the RL agent. By explicitly avoiding modeling task-irrelevant observation components, the latent dynamics should also become simpler and more sample-efficient to learn than the original (more complex, higher variance) dynamics on unfiltered observations. In simulations, ground-truth masks of relevant observation components are often easily accessible, for example, in MuJoCo (Todorov et al., 2012), through added calls to the simulator API. We term the method trained with our proposed replacement auxiliary task as Segmentation Dreamer (SD) and call the version trained with ground-truth masks SD$^{\text{GT}}$.

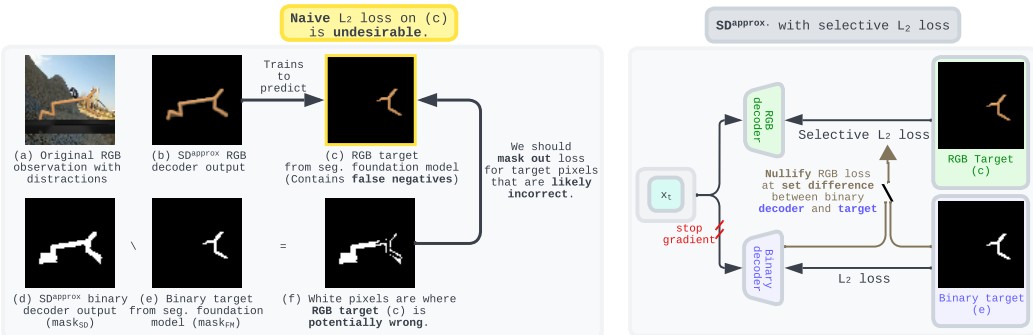

Figure 2: **Filtering $L_2$ loss to avoid training on false negatives in RGB labels.** *Left*: Estimated pixel locations (f) where the RGB target (c) is likely incorrectly masked out by the segmentation model (e). *Right*: A world model equipped with two decoders, one for reconstructing task-relevant masked RGB images and the other for binary masks, the targets for which are generated by a segmentation model. RGB $L_2$ loss is selectively masked by the set difference between (d) and (e). Latent representations ($x_t$) in the world model are subjected to the training signal only from the RGB branch. The binary branch is only utilized for selective $L_2$ loss.

## 4.2 LEVERAGING APPROXIMATE SEGMENTATION MASKS

A simulator capable of providing ground-truth masks for task-relevant regions is not always available. For such cases where only RGB images are available from the environment, we propose to fine-tune a segmentation mask foundation model to our domain and integrate its predictions into the SD training pipeline. Below, we describe our method for training with approximate task-relevance masks, termed $\text{SD}^{\text{approx.}}$.

As an offline process before training the world model, we fine-tune a segmentation model with a small number of example RGB images and their segmentation masks annotations that indicate task-relevant image regions. Thanks to recent advances in segmentation foundation models, we can obtain a new domain-specific mask model with a very small amount of training examples. For our experiments, we use the Personalized SAM (PerSAM) (Zhang et al., 2023) using one-shot adaptation and SegFormer (Xie et al., 2021) fine-tuned with 5 and 10 examples. For the sake of controlled and reproducible evaluation, we extract these RGB and mask training pairs from simulators, however, this number of required samples is small enough that it can be collected with expert human annotation as well. Also, although we use these specific foundation models, our method should also be compatible with *any* semantic masking method. Further details such as how we obtain the fine-tuning data can be found in Appendix K. Once fine-tuning is complete, we incorporate the segmentation model into the SD pipeline to create pseudo-labels for our proposed auxiliary task.

## 4.3 LEARNING IN THE PRESENCE OF MASKING ERRORS

Although foundation segmentation models generalize well to new scenarios (e.g., different poses, occlusions), prediction errors are inevitable (Fig. 1d). Since each frame is processed independently, segmentation predictions can flicker along trajectories. False negatives in task relevance are particularly detrimental when using naive $L_2$ loss on image reconstruction. Missing relevant scene elements in reconstruction targets can lead the encoder to learn incomplete representations, dropping essential task-related information. This variability disrupts the learning of accurate representations and dynamics in the world model.

Despite noisy targets, neural networks can self-correct if most labels are accurate (Han et al., 2018). Additionally, DREAMER's use of GRUs (Cho et al., 2014) provides temporal consistency even with flickering targets. However, as illustrated in Fig. 2 (b)&(c), it's undesirable to propagate gradients from regions where the original image has been incorrectly masked out. Allowing gradients from these regions provides misleading signals. If we could identify the incorrect regions in the reconstruction target, we could nullify the decoder's $L_2$ loss there—a technique we call selective $L_2$ loss.

Since we cannot directly identify regions where the RGB target is incorrectly masked due to false negatives, we estimate them. Preliminary experiments show that a binary mask decoder from world

model states (as an added auxiliary task) can be less prone to transient false negatives, unlike RGB prediction, which tends to memorize noisy labels. Therefore, we propose training a world model with two reconstruction tasks (Fig. 2, right): one decoding masked RGB images and the other predicting task-relevance binary masks. Both use the foundation model's binary mask, $\text{mask}_{\text{FM}}$, to construct targets. The RGB branch decodes masked RGB images, while the binary branch predicts $\text{mask}_{\text{FM}}$. We denote the binary masks produced by the world model as $\text{mask}_{\text{SD}}$, where pixels labeled *true* indicate task relevance.

To avoid training on incorrectly masked-out regions, we estimate where $\text{mask}_{\text{FM}}$ may be falsely negative by finding disagreements with $\text{mask}_{\text{SD}}$. Specifically, we selectively nullify RGB decoder $L_2$ loss for regions marked false in $\text{mask}_{\text{FM}}$ but predicted true in $\text{mask}_{\text{SD}}$. This prevents training on potentially falsely masked-out pixels still considered task-relevant by a second predictor. Formally, the mask for selective $L_2$ loss is the set difference between true pixel locations in $\text{mask}_{\text{SD}}$ and $\text{mask}_{\text{FM}}$:

$$\text{pixel}_{\text{MaskOut}} = \text{pixel}_{\text{SD}} \setminus \text{pixel}_{\text{FM}} \tag{6}$$

where $\text{pixel}_{\text{MaskOut}}$ indicates pixels to nullify loss at, and $\text{pixel}_{\text{SD}}$ and $\text{pixel}_{\text{FM}}$ are pixels marked true in $\text{mask}_{\text{SD}}$ and $\text{mask}_{\text{FM}}$, respectively.

Fig. 2 (d–f) shows examples of $\text{mask}_{\text{SD}}$, $\text{mask}_{\text{FM}}$, and $\text{pixel}_{\text{MaskOut}}$. See Appendix L for details on obtaining $\text{mask}_{\text{SD}}$. Our experiments indicate that selective $L_2$ loss effectively overcomes noisy segmentation labels and improves downstream agent performance.

Lastly, we observe better performance when we prevent gradients from the binary mask decoding objective from propagating into the world model, so we apply a stop gradient to the inputs of the mask decoder head (see Appendix G for ablations).

## 5 EXPERIMENTS

We evaluate our method on a variety of visual robotic control tasks from the DeepMind Control Suite (DMC) (Tassa et al., 2018) and Meta-World (Yu et al., 2019). Since the standard environments in these benchmarks have simple backgrounds with minimal distractions, we introduce visual distractions by replacing the backgrounds with random videos from the 'driving car' class in the Kinetics 400 dataset (Kay et al., 2017), following prior work (Zhang et al., 2021; Nguyen et al., 2021; Deng et al., 2022). Details about the environment setup and task visualizations are provided in Appendices H and B. In evaluation, we roll out policies over 10 episodes and compute the average episode return. Unless otherwise specified, we report the mean and standard error of the mean (SEM) of four independent runs with different random seeds. We use default DREAMER-V3 hyperparameters in all experiments.

### 5.1 DMC EXPERIMENTS

We evaluate SD on six tasks from DMC featuring different forms of contact dynamics, degrees of freedom, and reward sparsities. For each task, models are trained for 1M environment steps generated by 500K policy decision steps with an action repeat of 2.

#### 5.1.1 COMPARISON WITH DREAMER

We compare our methods, $\text{SD}^{\text{GT}}$ and $\text{SD}^{\text{approx.}}$, to the base DREAMER (Hafner et al., 2023) method. Here, $\text{SD}^{\text{approx.}}$ is denoted as $\text{SD}_N^{\text{FM}}$, specifying the segmentation model used (FM) and the number of fine-tuning examples ($N$). All methods are trained in distracting environments, except for the DREAMER* baseline, which is trained in the original environment without visual distractions. In most cases, we consider DREAMER* as an upper bound for methods trained with distractions. Similarly, $\text{SD}^{\text{GT}}$ serves as an upper bound for $\text{SD}^{\text{approx.}}$, with the performance gap expected to decrease in the future as segmentation quality improves.

As shown in Fig. 3a, DREAMER fails across all tasks due to task-irrelevant information in RGB reconstruction targets, which wastes latent capacity and complicates dynamics learning. In contrast, $\text{SD}^{\text{GT}}$ achieves test returns comparable to DREAMER* by focusing on reconstructing essential features and ignoring irrelevant components. Interestingly, $\text{SD}^{\text{GT}}$ outperforms DREAMER* in Cartpole

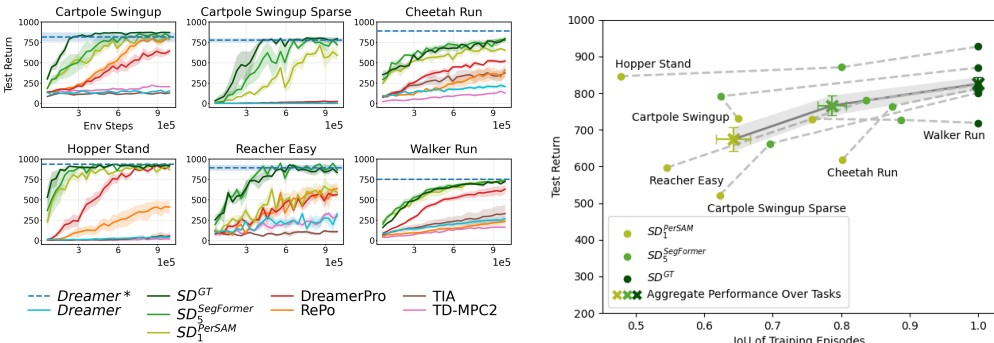

(a) Environment Steps vs. Expected Test Return     (b) IoU during Training vs. Expected Test Return

Figure 3: (a) **Learning curves on six visual control tasks from DMC.** Every method but DREAMER* is trained on distracting environments. All curves show the mean over 4 seeds with the standard error of the mean (SEM) shaded. (b) Segmentation quality during training vs. downstream task performance. Best viewed in color.

Swingup, possibly because the original environment still contains small distractions (e.g., moving dots) that DREAMER* has to model.

A limitation of SD is its reliance on acccurate and correct prior knowledge to select task-relevant components. In Cheetah Run, $SD^{GT}$ underperforms compared to DREAMER*, likely because we only include the cheetah's body in the mask, excluding the ground plate, which may be important for contact dynamics. Visual examples and further experiments are in Appendices B and C.

For $SD^{approx.}$, we test with two foundation models: PerSAM adapted with one RGB example and its GT mask, and SegFormer adapted with five such examples. Despite slower convergence due to noisier targets, both $SD_1^{PerSAM}$ and $SD_5^{SegFormer}$ achieve similar final performance to $SD^{GT}$ in most tasks. A failure case for $SD_1^{PerSAM}$ is Reacher Easy, where a single data point is insufficient to obtain a quality segmentation for the small task-relevant objects.

### 5.1.2 COMPARISON WITH BASELINES

We compare **$SD^{approx.}$** with state-of-the-art methods, including DreamerPro (Deng et al., 2022), RePo (Zhu et al., 2023), TIA (Fu et al., 2021), and TD-MPC2 (Hansen et al., 2023). DreamerPro incorporates prototypical representation learning in the DREAMER framework; RePo minimizes mutual information between observations and latent states while maximizing it between states and future rewards; TIA learns separate task-relevant and task-irrelevant representations which can be combined to decode observations; and TD-MPC2 decodes a terminal value function. Among these baselines, only TIA relies on observation reconstruction. Further details are in Appendix M.

Our results in Fig. 3a show that our method consistently outperforms the baselines in performance and sample efficiency. TIA underperforms in many tasks, requiring many samples to infer task-relevant observations from rewards and needing exhaustive hyperparameter tuning. Even with optimal settings, it may lead to degenerate solutions where a single branch captures all information. In contrast, our method focuses on task-relevant parts without additional tuning by effectively injecting prior knowledge. RePo performs comparably to ours in Cartpole Swingup but underperforms in other tasks and converges more slowly.

TD-MPC2 struggles significantly in distracting environments. We speculate that spurious correlations from distractions introduce noise to value-function credit assignment that hinders representation learning. Our method mitigates this by directly supervising task-relevant features using segmentation models, leading to more consistent and lower-variance targets.

Among these methods, DreamerPro is the most competitive, demonstrating the effectiveness of prototypical representation learning for control. However, it often requires more environment interactions and converges to lower performance.

In the Cartpole Swingup with sparse rewards, none of the prior works successfully solved the task, highlighting the challenge of inferring task relevance from weak signals. Our method achieves near-

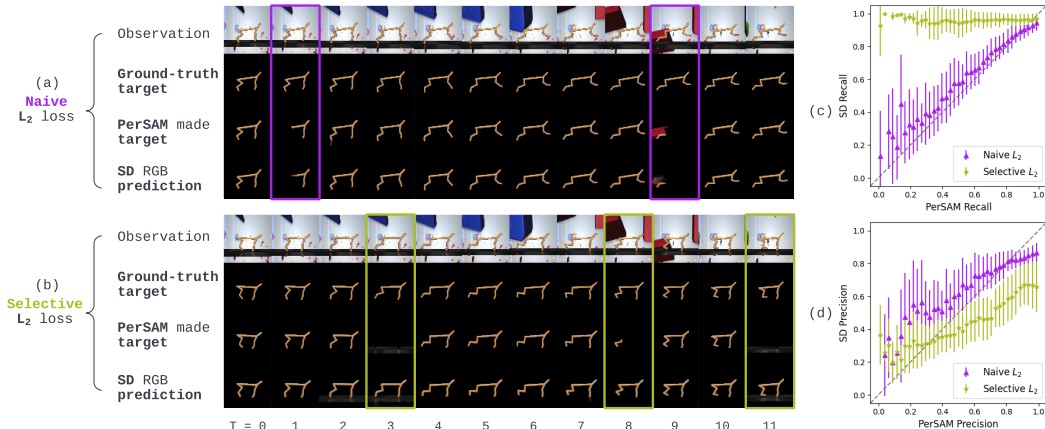

Figure 4: (a)+(b) **Qualitative comparison of SD trained with naive and selective $L_2$ loss.** Trajectories are taken from each method's train-time replay buffer, selected to have the same background. Frames with PerSAM error are highlighted. The model trained with the selective $L_2$ loss overcomes errors in the target, whereas the one trained with the naive $L_2$ loss memorizes target errors. (c)+(d) shows the precision and recall of PerSAM and the SD RGB decoder prediction. SD RGB predictions are binarized using a threshold to compute recall and precision w.r.t. the ground-truth mask. The data points used for plotting are from the same Cheetah Run training experiment as in (a)+(b). The selective $L_2$ loss significantly improves the recall with only a moderate impact on precision.

oracle performance, being the only one to train an agent with sparse rewards amidst distractions. This suggests the potential to train agents in real-world, distraction-rich environments without extensive reward engineering.

### 5.1.3 ABLATION STUDY

We investigate the effects of the components in SD$^{\text{approx.}}$ by addressing: (1) the benefits of using segmentation models for targets vs. input preprocessing; (2) the effectiveness of the selective $L_2$ loss compared to the naive $L_2$ loss; and (3) the impact of the segmentation quality on RL performance. In these experiments, we fine-tune PerSAM with a single data point for segmentation mask prediction.

**Using segmentation masks for an auxiliary task vs. input preprocessing.** We create a variant of SD$_1^{\text{PerSAM}}$ that uses masked observations for both inputs and targets, denoted in Tab. 1 by *As Input*. These results suggest that SD$_1^{\text{PerSAM}}$, in addition to not requiring mask prediction at test-time, also achieves better test performance and lower variance. Using predicted masks as input is more prone to segmentation errors, restricting the agent's perception when masks are incorrect and making training

Table 1: Final performance of SD variants. Mean over 4 runs with the standard error of the mean is reported. The highest means are highlighted.

| Task | SD$_1^{\text{PerSAM}}$ | As Input | Naive $L_2$ |
|---|---|---|---|
| Cartpole Swingup | **730 ± 75** | 565 ± 108 | 719 ± 62 |
| Cartpole Swingup Sparse | **521 ± 92** | 457 ± 151 | 408 ± 114 |
| Cheetah Run | **619 ± 35** | 524 ± 37 | 486 ± 58 |
| Hopper Stand | **846 ± 27** | 689 ± 39 | 790 ± 51 |
| Reacher Easy | 597 ± 97 | **642 ± 116** | 415 ± 50 |
| Walker Run | **730 ± 13** | 589 ± 28 | 557 ± 51 |

more challenging. In contrast, SD$^{\text{approx.}}$ receives intact observations, with task-relevant filtering at the encoder level, leading to better state abstraction. Further analysis on test-time segmentation quality's impact is in Appendix D.

**Selective $L_2$ loss vs. naive $L_2$ loss.** As shown in Tab. 1, SD$_1^{\text{PerSAM}}$ consistently outperforms the *Naive $L_2$* variant, especially in complex tasks like Cheetah Run and Walker Run. Segmentation models often miss embodiment components (Fig. 4, third row). With the naive $L_2$ loss, the model replicates these errors, leading to incomplete latent representations and harming dynamics learning (Fig. 4a, fourth row). In contrast, SD$^{\text{approx.}}$ self-corrects by skipping the $L_2$ computation where PerSAM targets are likely wrong (Fig. 4b, fourth row). Fig. 4(c)&(d) show that the naive $L_2$ loss

follows PerSAM's trends, while the selective $L_2$ loss recovers from poor recall with only a moderate precision decrease.

**Impact of segmentation quality on RL performance.** Fig. 3b plots the training-time segmentation quality against the RL agent's test-time performance. Comparing three SD variants with different mask qualities (two estimated, one ground truth), we observe that better segmentation tends to lead to higher RL performance, as accurate targets better highlight task-relevant components. This suggests that improved segmentation models can enhance agent performance without ground-truth masks. In Cartpole Swingup, one of two exceptions, the IoU difference between $SD_1^{PerSAM}$ and $SD_5^{SegFormer}$ is small, and the test returns may fall within the margin of error. In Walker Run, the other exception, all variants show high segmentation quality and reach near-optimal performance. Here, we hypothesis that a small amount of noise in the target may act as a regularizer, contributing to marginally better downstream performance.

## 5.2 META-WORLD EXPERIMENTS

Object manipulation is a natural application for our method where prior knowledge can be applied straightforwardly by identifying and masking task-relevant objects and robot embodiments. We evaluate SD on six tasks from Meta-World (Yu et al., 2019), a popular benchmark for robotic manipulation. Depending on the difficulty of each task, we conduct experiments for 30K, 100K, and 1M environment steps, with an action repeat of 2 (details in Appendix I). Preliminary tests showed that Seg-Former performs well with few-shot learning on small objects. We fine-tune SegFormer with 10 data points to estimate masks in these experiments.

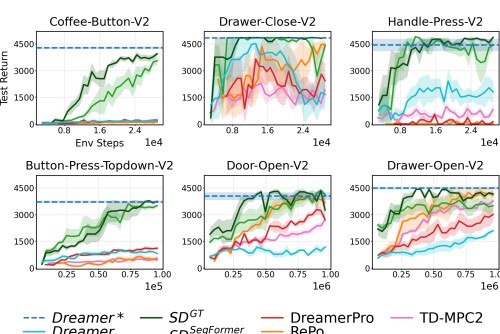

Figure 5: **Learning curves on six visual robotic manipulation tasks from Meta-World.** All curves show the mean over 4 seeds with the standard error of the mean shaded.

Fig. 5 suggests that our approach outperforms the baselines overall, with a more pronounced advantage in tasks involving small objects like Coffee-Button. Our method excels because it focuses on small, task-relevant objects, avoiding the reconstruction of unnecessary regions that occupy much of the input. In contrast, the baselines struggle as they often underestimate the significance of these small yet highly task-relevant objects. Among the baselines, RePo (Zhu et al., 2023) is the most competitive. However, RePo performs poorly in a sparse reward setup (see Appendix J).

## 6 CONCLUSION

In this paper, we propose SD, a simple yet effective method for learning task-relevant features in MBRL frameworks like DREAMER by using segmentation masks informed by domain knowledge. Using ground-truth masks, $SD^{GT}$ achieves performance comparable with undistracted DREAMER with high sample efficiency in distracting environments when provided with accurate prior knowledge. Our main method, $SD^{approx.}$, uses mask estimates from off-the-shelf one-shot or few-shot segmentation models and employs a selective $L_2$ loss. It learns effective world models that produce strong agents outperforming baselines.

To the best of our knowledge, our approach appears to be the first model-based approach to successfully train an agent in a sparse reward environment under visual distractions, enabling robust agent training without extensive reward engineering. This work also advances the integration of computer vision and RL by presenting a novel way to leverage recent advances in segmentation to address challenges in visual control tasks. The proposed method achieves strong performance on diverse tasks with distractions and effectively incorporates human input to indicate task relevance. This enables practitioners to readily train an agent for their own purposes without extensive reward engineering. However, SD has some limitations to consider in future work, which we further explore in Appendix O.

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

# A CODE RELEASE

We plan to make the code for Segmentation Dreamer publicly available upon acceptance.

# B VISUALIZATION OF TASKS

## B.1 DEEPMIND CONTROL SUITE (DMC)

Fig. 6 visualizes the six tasks in DMC (Tassa et al., 2018) used in our experiments. Each row presents the observation from the standard environment, the corresponding observation with added distractions, the ground-truth segmentation mask, and the RGB target with the ground-truth mask applied. Cartpole Swingup Sparse and Cartpole Swingup share the same embodiment and dynamics. Cartpole Swingup Sparse only provides a reward when the pole is upright, whereas Cartpole Swingup continuously provides dense rewards weighted by the proximity of the pole to the upright position. Reacher Easy entails two objects marked with different colors in the segmentation mask, as shown in Fig. 6e 3rd column. Before passing the mask to SD, the mask is converted to a binary format where both objects are marked as *true* as task-relevant.

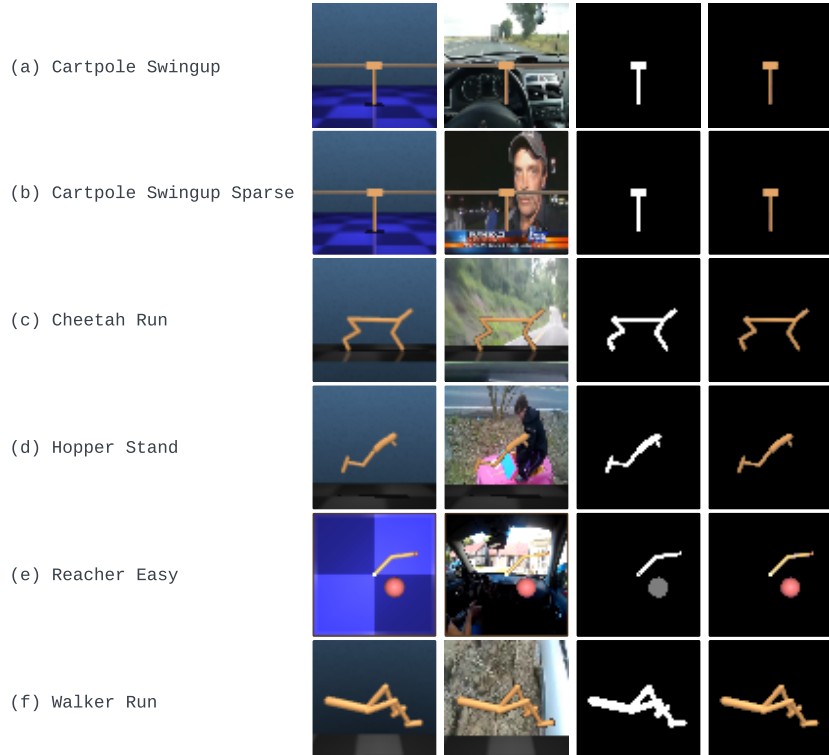

Figure 6: DMC tasks. Left to right: (1) standard environment observations, (2) distracting environment observations, (3) ground-truth segmentation masks, and (4) RGB observations with ground-truth masks applied. We use (4) as auxiliary reconstruction targets in $SD^{GT}$.

## B.2 META-WORLD

Fig. 7 shows the six tasks from Meta-World-V2 used in our experiments. Meta-World is a realistic robotic manipulation benchmark with challenges such as multi-object interactions, small objects, and occlusions.

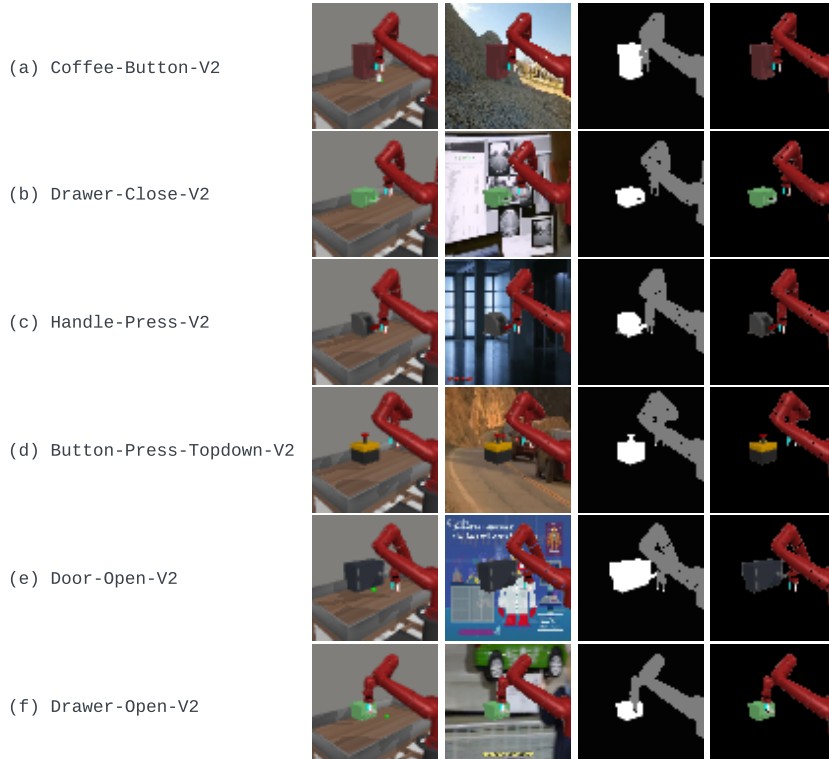

(a) Coffee-Button-V2

(b) Drawer-Close-V2

(c) Handle-Press-V2

(d) Button-Press-Topdown-V2

(e) Door-Open-V2

(f) Drawer-Open-V2

Figure 7: Meta-World tasks. Left to right: (1) standard environment observations, (2) distracting environment observations, (3) ground-truth segmentation masks, and (4) RGB observations with ground-truth masks applied. We use (4) as auxiliary reconstruction targets in $SD^{GT}$. Masks with multiple classes for different objects are converted to binary masks (all non-background regions are *true* and task-relevant) before use with SD.

## C  THE IMPACT OF PRIOR KNOWLEDGE

We investigate the impact of accurate prior knowledge of task-relevant objects. Specifically, we conduct additional experiments on Cheetah Run—the task showing the largest disparity between DREAMER* and SD$^{GT}$ in Fig. 3a. In our primary experiment, we designated only the cheetah's body as the task-relevant object. However, since the cheetah's dynamics are influenced by ground contact, the ground plate should have also been considered task-relevant.

Fig. 8 (a–c) illustrates the observation with distractions, the auxiliary target without the ground plate, and with the ground plate included, respectively. Fig. 8d compares SD$^{GT}$ trained with different selections of task-relevant objects included in the masked RGB reconstruction targets. We show that including the ground plate leads to faster learning and performance closer to that of the oracle. This highlights the significant influence of prior knowledge on downstream tasks, suggesting that comprehensively including task-relevant objects yields greater benefits.

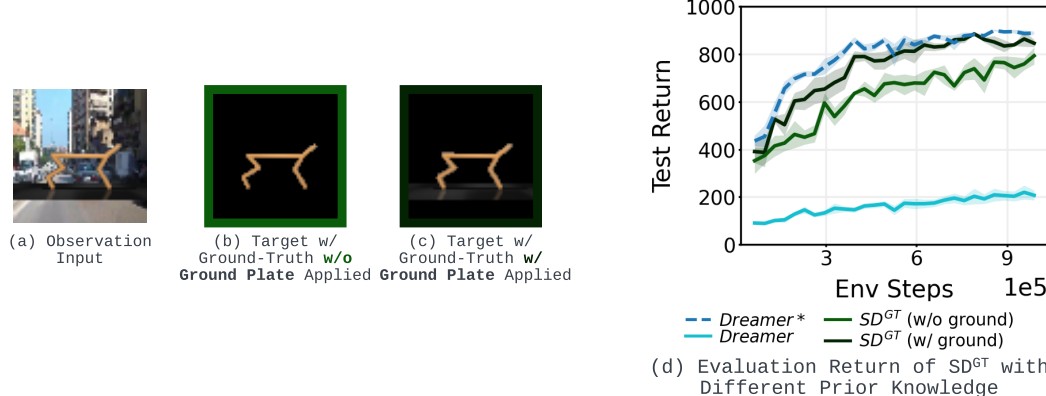

(a) Observation Input

(b) Target w/ Ground-Truth **w/o** **Ground Plate** Applied

(c) Target w/ Ground-Truth **w/** **Ground Plate** Applied

(d) Evaluation Return of SD$^{GT}$ with Different Prior Knowledge

Figure 8: **The impact of prior knowledge on Cheetah Run.** (d) The mean over 4 seeds with the standard error of the mean (SEM) is shaded.

# D  THE IMPACT OF TEST-TIME SEGMENTATION QUALITY ON PERFORMANCE

We investigate how test-time segmentation quality affects $SD^{approx.}$ as well as the *As Input* variation that applies mask predictions to RGB inputs in addition to reconstruction targets. For this analysis, we use PerSAM fine-tuned with a single data point for segmentation prediction. To measure segmentation quality, we compute episodic segmentation quality by averaging over frame-level IoU. In Fig. 9 we plot episode segmentation quality versus test-time reward on the evaluation episodes during the last 10% of training time.

Fig. 9 illustrates that $SD^{approx.}$ exhibits greater robustness to test-time segmentation quality compared to the *As Input* variation, with the discrepancy increasing as the IoU decreases. This disparity primarily arises because *As Input* relies on observations restricted by segmentation predictions, and thus its performance deteriorates quickly as the segmentation quality decreases. In contrast, $SD^{approx.}$ takes the original observation as input and all feature extraction is handled by the observation encoder, informed by our masked RGB reconstruction objective. Consequently, $SD^{approx.}$ maintains resilience to test-time segmentation quality.

An intriguing observation is that a poorly trained agent can lead to poor test-time segmentation quality. For instance, Cartpole Swingup (Sparse) exhibits different segmentation quality distributions between $SD^{approx.}$ and *As Input*. This discrepancy occurs because the sub-optimal agent often positions the pole at the cart track edge, causing occlusion and hindering accurate segmentation prediction by PerSAM.

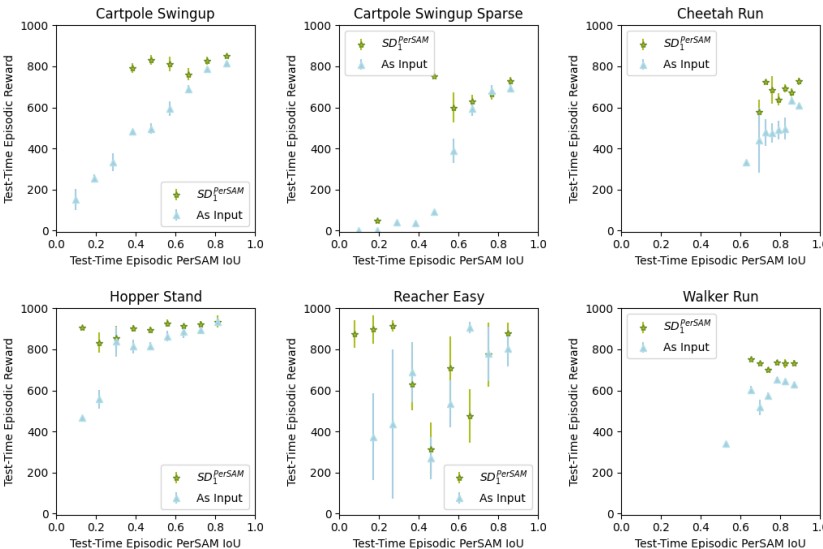

Figure 9: Test-time episodic reward vs PerSAM episodic IoU for $SD_1^{PerSAM}$ and *As Input* ($SD_1^{PerSAM}$ with masked RGB observations as input). $SD_1^{PerSAM}$ is more robust to test-time segmentation prediction errors.

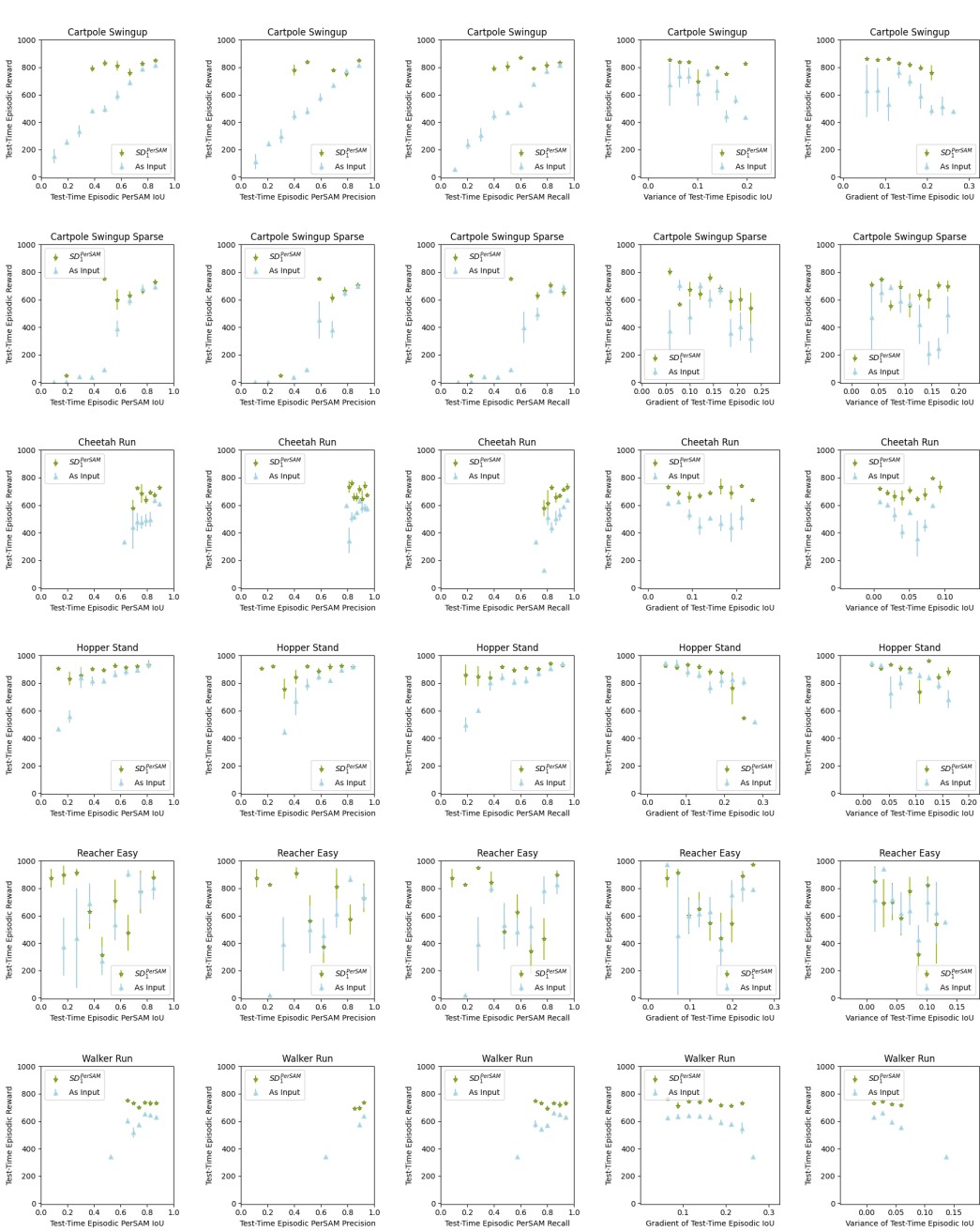

Figure 10: Test-time episodic reward plotted against IoU, precision, recall, IoU variance, and IoU gradient, respectively.

## E    EXPERIMENTS WITH DIFFERENT TYPES OF DISTRACTIONS

In this section, we investigate how our method performs when faced with types of distractions beyond background distractions. Specifically, we consider three additional types of distractions: foreground distractions, color changes in foreground objects, and camera angle perturbations. These experiments are conducted on the DeepMind Control Suite.

### E.1    EXPERIMENT SETUP

To ensure robustness against distractions during testing, we introduce domain randomizations during training. Specifically, both the segmentation models and SD are trained under domain-randomized environments. And, we evaluate our method on a distribution of perturbations that matches the variability introduced at training time.

**Foreground Distractions.**    To simulate distractions that occlude or block task-relevant parts of the scene, we introduce a moving foreground distractor. This is implemented as a blue rectangle rendered near the center of the scene for 4–6 frames every 18–22 frames. These intervals are uniformly sampled each time the distractor appears, meaning approximately 25% of the frames in an episode include the distractor. The distractor moves along pixel-space trajectories defined by randomized $\Delta x$ and $\Delta y$ values within the range of (-3,3), which are sampled each time the distractor appears. The goal of introducing this type of distractor is to assess whether our method remains robust in the presence of occlusions that can interfere with task-relevant visual information.

Although this preliminary setup simplifies the distractor's appearance and trajectory, it can easily be extended to incorporate more complex objects or movement patterns. Given the capabilities of visual foundation models (VFMs), we hypothesize that our method will generalize well to a variety of foreground distractors with different properties.

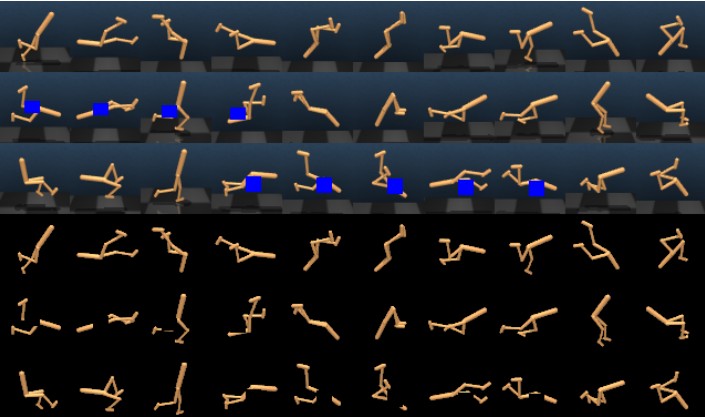

Figure 11: Examples of **foreground distractors** in the environment and corresponding **predictions** from the **segmentation** model that remains **robust** to occlusions in the test set.

**Color Changes in Foreground Objects.**    For color perturbations, we simulate changes in the appearance of the agent or task-relevant objects. Following the approach of Stone et al. (2021), we apply a max delta of 0.1 and set step std to 0.0, resulting in a static color throughout the episode. These changes simulate environmental factors such as lighting variations that may occur during deployment. This experiment evaluates the ability of the model to adapt to changes in the visual characteristics of task-critical elements.

**Camera Angle Perturbations.**    To introduce changes in camera perspective, we follow the implementation of Stone et al. (2021) and apply a scaling factor of 0.1. This results in shifts in the camera view, which simulate real-world deployment scenarios where the agent's viewpoint may vary due to physical movement or environmental adjustments. These perturbations test the model's capacity to maintain performance under altered visual perspectives, as illustrated in Fig. 13.

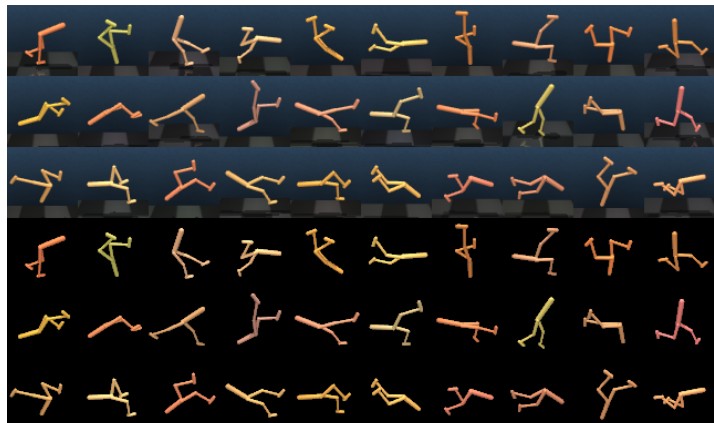

Figure 12: Examples of **color** perturbations applied to the agent and corresponding predictions from the segmentation model that remains robust to color changes in the test set.

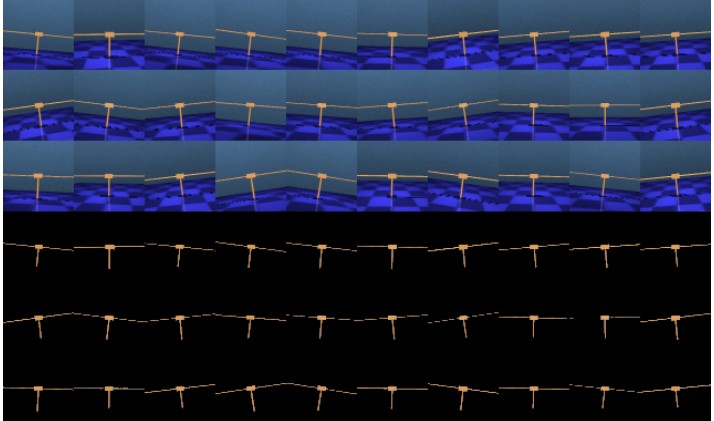

Figure 13: Examples of **camera view** perturbations and corresponding predictions from the segmentation model that remains robust to camera view variations in the test set.

**Background Distractions.** See Fig. 14 for examples of background perturbations.

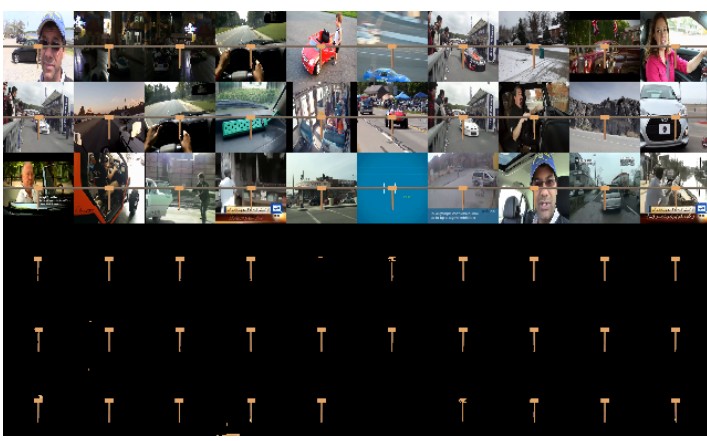

Figure 14: Examples of **background distractions** and predictions from the segmentation model in the test set.

**Background and Color Perturbation.**    See Fig. 15 for examples of background and color perturbations.

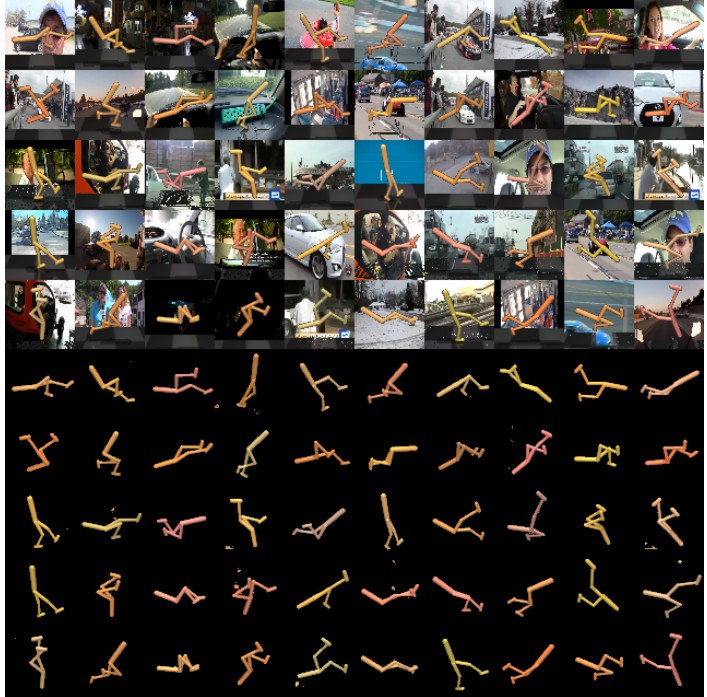

Figure 15: Examples of **background distractions** and **color** perturbations and predictions from the segmentation model in the test set.

## F Segmentation Quality in Meta-World

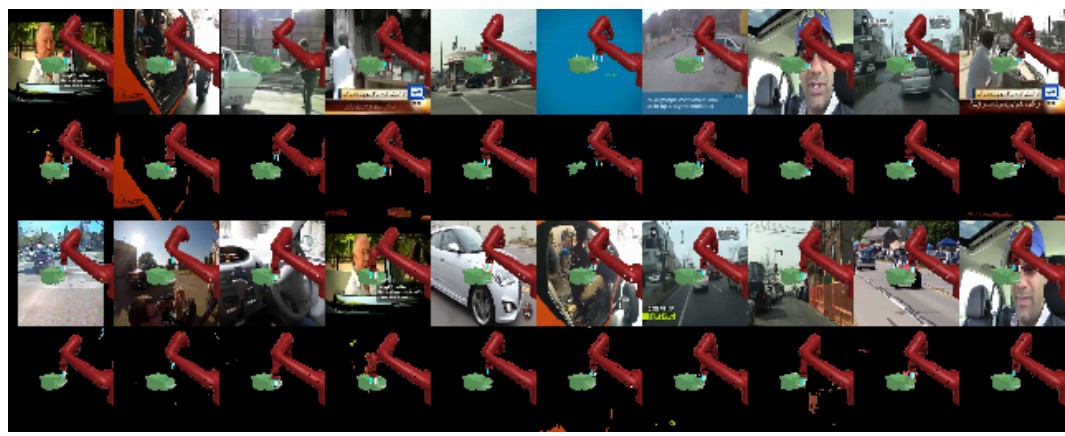

Figure 16: Examples of background perturbations and corresponding **predictions** from the **segmentation** model on Drawer-Open-V2 in the test set.

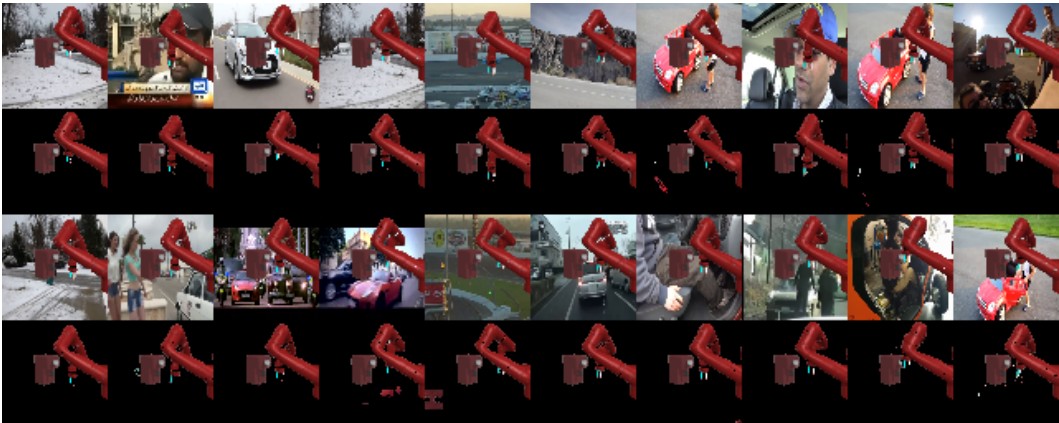

Figure 17: Examples of background perturbations and corresponding **predictions** from the **segmentation** model on Coffee-Button-V2 in the test set.

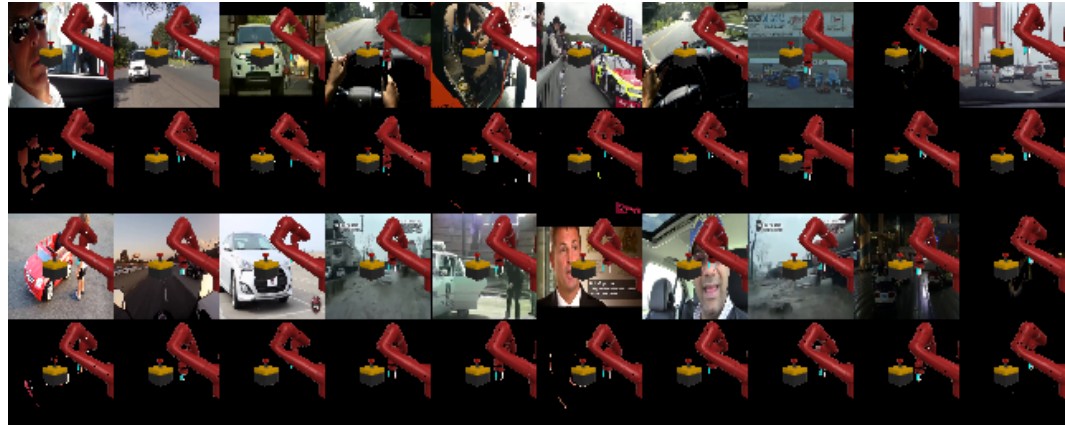

Figure 18: Examples of background perturbations and corresponding **predictions** from the **segmentation** model on Button-Press-Topdown-V2 in the test set.

# G    ABLATION WITHOUT STOP GRADIENT

**Should the SD$^{\text{approx.}}$ world model be shielded from gradients of the binary mask decoder head?**

To estimate potential regions on RGB targets where task-relevant regions are incorrectly masked out, we train a binary mask prediction head on the world model to help detect false negatives in masks provided by the foundation model. We see better performance when gradients from this binary mask decoder objective are not propagated to the rest of the world model. Thus, the default SD$^{\text{approx.}}$ architecture is trained with the gradients of the binary mask branch stopped at its $[h_t; z_t]$ inputs, and the latent representations in the world model are trained only by the task-relevant RGB branch in addition to the standard DREAMER reward/continue prediction and KL-divergence between the dynamics prior and observation encoder posterior. Tab. 2 shows that the performance drops significantly when training without stopping these gradients.

We also examine masks predicted by the binary mask decoder head in Fig. 19. Predictions are coarser grained than their RGB counterparts, lacking details important for predicting intricate forward dynamics. Overall, reconstructing RGB observations with task-relevance masks applied demonstrates itself as a superior inductive bias to learn useful features for downstream tasks compared to binary masks or raw unfiltered RGB observations.

Table 2: Final performance of SD and SD without stop gradient.

| Task | SD$_1^{\text{PerSAM}}$ | No SG |
|---|---|---|
| Cartpole Swingup | **730 ± 75** | 439 ± 81 |
| Cartpole Swingup Sparse | **521 ± 92** | 112 ± 40 |
| Cheetah Run | **619 ± 35** | 376 ± 50 |
| Hopper Stand | **846 ± 27** | 587 ± 127 |
| Reacher Easy | **597 ± 97** | 273 ± 74 |
| Walker Run | **730 ± 13** | 407 ± 62 |

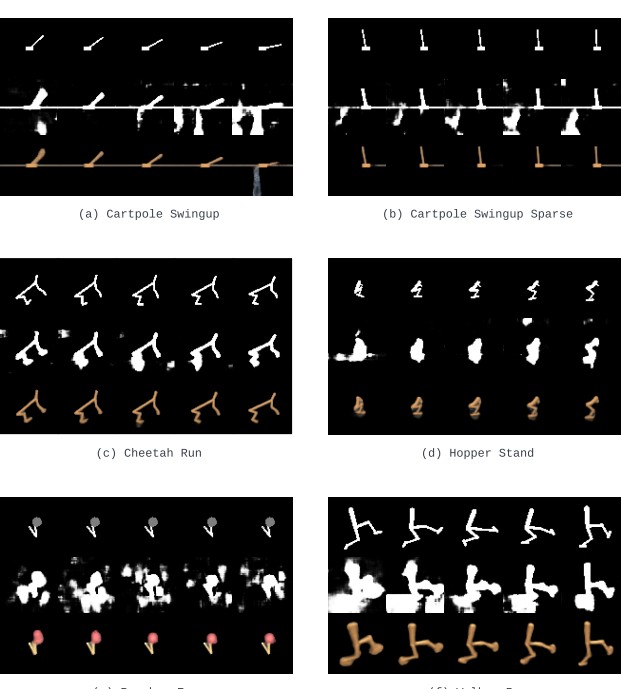

(a) Cartpole Swingup

(b) Cartpole Swingup Sparse

(c) Cheetah Run

(d) Hopper Stand

(e) Reacher Easy

(f) Walker Run

Figure 19: From the top row to the bottom row: (1) ground-truth segmentation masks, (2) SD$^{\text{approx.}}$ binary mask predictions, and (3) SD$^{\text{approx.}}$ RGB predictions.

## H  DISTRACTING DMC SETUP

We follow the DBC (Zhang et al., 2021) implementation to replace the background with color videos. The ground plate is also presented in the distracting environment. We used hold-out videos as background for testing. We sampled 100 videos for training from the Kinetics 400 training set of the 'driving car' class, and test-time videos were sampled from the validation set of the same class.

## I  DISTRACTING META-WORLD SETUP

We test on six tasks from Meta-World-V2. For all tasks, we use the `corner3` camera viewpoint. The maximum episode length for Meta-World tasks is 500 environment steps, with the action repeat of 2 (making 250 policy decision steps). We classify these tasks into `easy`, `medium`, and `difficult` categories based on the training curve of DREAMER* (DREAMER trained in the standard environments). Coffee Button, Drawer Close, and Handle Press are classified as `easy`, and we train baselines on these for 30K environment steps. Button Press Topdown (`medium`) is trained for 100K steps, and Door Open and Drawer Open (`difficult`) are trained for 1M environment steps.

## J  RESULTS ON META-WORLD WITH SPARSE REWARDS

We also evaluate on sparse reward variations of the distracting Meta-World environments where a reward of 1 is only provided on timesteps when a *success* signal is given by the environment (e.g. objects are at their goal configuration). Rewards are 0 in all other timesteps. The maximum attainable episode reward is 250.

The sparse reward setting is more challenging because the less informative reward signal makes credit assignment more difficult for the RL agent. Fig. 20 shows that our method consistently achieves higher sample efficiency and better performance, showing promise for training agents robust to visual distractions without extensive reward engineering. In Meta-World experiments, TIA (Fu et al., 2021) is not included as it requires exhaustive hyperparameter tuning for new domains and is the lowest-performing method in DMC in general.

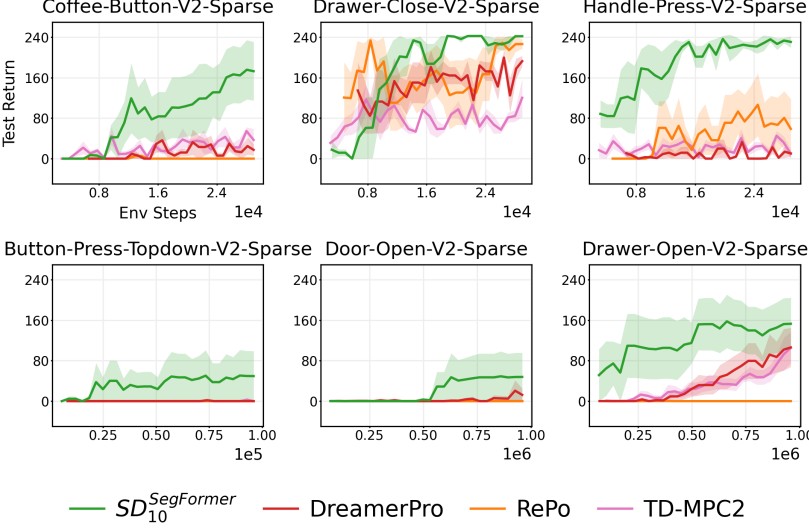

Figure 20: Learning curves on six visual robotic manipulation tasks from Meta-World with sparse rewards.

## K   FINE-TUNING PERSAM AND SEGFORMER

In this section, we describe how we fine-tune segmentation models and collect RGB and segmentation mask examples to adapt them.

**PerSAM.** Personalized SAM (PerSAM) (Zhang et al., 2023) is a segmentation model designed for personalized object segmentation building upon the Segment Anything Model (SAM) (Kirillov et al., 2023). This model is particularly a good fit for our SD use case since it can obtain a personalized segmentation model without additional training by one-shot adapting to a *single* in-domain image. In our experiments, we use the model with ViT-T as a backbone.

**SegFormer.** We use 5 or 10 pairs of examples to fine-tune SegFormer (Xie et al., 2021) MiT-b0.

To collect a one-shot in-domain RGB image and mask example for DMC and MetaWorld experiments, we sample a state from the initial distribution $p_0$ and render the RGB observation. In a few-shot scenario, we deploy a random agent in to collect more diverse observations from more diverse states.

To generate the associated masks for these states, we make additional queries to the simulation rendering API. We represent the pixel values for background and irrelevant objects as *false* and task-relevant objects as *true*. In multi-object cases, we may perform a separate adaptation operation for each task-relevant object, resulting in more than 2 mask classes. In such cases, before integrating masks with $SD^{approx.}$, we will combine the union of the mask classes for all pertinent objects as a single *true* task-relevant class, creating a binary segmentation mask compatible with our method.

In cases where example masks cannot be programmatically extracted, because such a small number of examples are required (1-10), it should also be very feasible for a human to use software to manually annotate the needed mask examples from collected RGB images.

## L   DETAILS ON SELECTIVE $L_2$ LOSS

The binary mask prediction branch in $SD^{approx.}$ is equipped with the sigmoid layer at its output. In order to obtain binary $mask_{SD}$, we binarize the SD binary mask prediction with a threshold of 0.9.

## M   DETAILS ON BASELINES

It is known that RePo (Zhu et al., 2023) outperforms many earlier works (Fu et al., 2021; Hansen et al., 2022; Zhang et al., 2021; Wang et al., 2022; Gelada et al., 2019) and that DreamerPro (Deng et al., 2022) surpasses TPC (Nguyen et al., 2021). However, theses two groups of works have been using slightly different environment setups and have not been compared with each other despite addressing the same high-level problem on the same DMC environments. In our experiments, we evaluate the representatives in each cluster on a common ground (See Appendix H) and compare them with our method.

In our experiments, we use hyperparameters used in the original papers for all the baselines, except RePo (Zhu et al., 2023) in Meta-World. RePo does not have experiments on Meta-World in which case we use hyperparameters used for Maniskill2 (Gu et al., 2023) which is another robot manipulation benchmark.

## N   EXTENDED RELATED WORK

There are several model-based RL approaches which introduce new auxiliary tasks. Dynalang (Lin et al., 2024) integrates language modeling as a self-supervised learning objective in world-model training. It shows impressive performance on benchmarks where the dynamics can be effectively described in natural language. However, it is not trivial to apply this method in low-level control scenarios such as locomotion control in DMC. Informed POMDP (Lambrechts et al., 2024) introduces an information decoder which uses priviledged simulator information to decode a sufficient statistic for optimal control. This shares an idea of using additional information available at training time with our method $SD^{GT}$. Although this can be effective on training in simulation where well-shaped

proprioceptive states exist, it cannot be applied to cases where such information is hard to obtain. In goal-conditioned RL, GAP (Nair et al., 2020) proposed to decode the difference between the future state and the goal state to help learn goal-relevant features in the state space.

## O  LIMITATIONS

Segmentation Dreamer achieves excellent performance across diverse tasks in the presence of distractions and provides a human interface to indicate task relevance. This capability enables practitioners to readily train an agent for their specific purposes without suffering from poor learning performance due to visual distractions. However, there are several limitations to consider.

First, since $SD^{approx.}$ harnesses a segmentation model, it can become confused when a scene contains distractor objects that resemble task-relevant objects. This challenge can be mitigated by combining our method with approaches such as InfoPower (Bharadhwaj et al., 2022), which learns controllable representations through empowerment (Mohamed & Jimenez Rezende, 2015). This integration would help distinguish controllable task-relevant objects from those with similar appearances but move without agent interaction.

Second, our method does not explicitly address randomization in the visual appearance of *task-relevant* objects, such as variations in brightness, illumination, or color. Two observations of the same internal state but with differently colored task-relevant objects may be guided toward different latent representations because our task-relevant "pixel-value" reconstruction loss forces them to be differentiated. Ideally, these observations should map to the same state abstraction since they exhibit similar behaviors in terms of the downstream task. Given that training with pixel-value perturbations on task-relevant objects is easier compared to dealing with dominating background distractors (Stone et al., 2021), our method is expected to manage such perturbations effectively without modifications. However, augmenting our approach with additional auxiliary tasks based on behavior similarity (Zhang et al., 2021) would further enhance representation learning and directly address this issue.

Finally, our approximation model faces scalability challenges when task-relevant objects constitute an open set. For instance, in autonomous driving scenarios, obstacles are task-relevant but cannot be explicitly specified. While our method serves as an effective solution when task-relevant objects are easily identifiable, complementary approaches should be considered when this assumption does not hold true.

