# OpenReview forum: "Make the Pertinent Salient: Task-Relevant Reconstruction for Visual Control with Distractions"
_ICLR.cc/2025/Conference — Submitted to ICLR 2025_

### Official Review · Reviewer_L53c · 2024-10-29

**Soundness:** 3
**Presentation:** 3
**Contribution:** 2
**Rating:** 5
**Confidence:** 4

**Summary:**

The paper presents Segmentation Dreamer (SD), which improves Model-Based Reinforcement Learning (MBRL) in visually distracting environments. It focuses on task-relevant components using segmentation masks, and achieves better sample efficiency and performance, especially in sparse reward settings and when exposed to significant visual distractions.

**Strengths:**

Originality: Propose selective L2 loss to handle not perfect segmentation input
Quality: A good approach to introduce segmentation model to help MBRL
Clarity: Visualizations easy to read
Significance: Deal with distraction in RL policy deployment, which is important

**Weaknesses:**

The main weakness is that proposed algorithm mainly tested with visual distraction with swapped 2D background shift. This set of benchmarks are surrogate to real domain shifts but the segmentation model directly aims to remove that. Thereby, it would be hard to generalize to other cases such as change of viewing angle, color, lighting, texture etc.

**Questions:**

Can you also compare against other methods also utilizing  segmentation model? e.g., using SAM to get relevant part and feed that to some RL algorithm?

Can you discuss how would some extension of the proposed method handle cases such as change of viewing angle, color, lighting, texture etc.?

---

> ### Author Response · Authors · 2024-11-25
>
> \[Q1] Comparison with Other Methods Utilizing Segmentation Models
>
> - Prior works \[1, 2] have indeed used segmentation models with task-specific prior knowledge in RL. However, these approaches utilize segmentation for preprocessing input observations, which can be unstable at test time. If the segmentation model fails during deployment—such as when encountering unfamiliar scenarios—it disrupts the input and causes the agent to malfunction. In contrast, our method leverages segmentation models as an auxiliary task, significantly improving robustness to segmentation errors. This advantage is evident in Table 1 and Figure 9, where the “**As Input**”  method represents these prior approaches, highlighting their vulnerability in untested environments. By using segmentation as an auxiliary task rather than a preprocessing step, our design avoids these pitfalls and ensures greater reliability and efficiency while addressing the same core problem.
>
> - References:
>
> - \[1] "Generalizable Visual Reinforcement Learning with Segment Anything Model." arXiv, 2024.
>
> - \[2] "Empowering Embodied Visual Tracking with Visual Foundation Models and Offline RL." European Conference on Computer Vision (ECCV), 2024.
>
> \[W1] Generalization to other domain shifts
>
> - Thank you for this thoughtful observation. We conducted additional experiments with three types of distractions: changes in camera angles, foreground occlusions, and color variations. Please refer to General Response 1 for further details.
>
> \[Q2] Extension of the Proposed Method to Handle Cases Like Viewing Angle, Color, Lighting, and Texture Changes
>
> - While our method demonstrates robustness under domain randomizations, such as variations in camera angles, foreground occlusions, and color changes, we acknowledge that more rigorous perturbations or combinations of distractions require further exploration. Enhancing invariance in these scenarios is a promising direction for future work. For example, decoding canonical representations \[7] as an additional auxiliary task could promote invariance to foreground perturbations. Similarly, incorporating a contrastive loss to map similar behaviors of different foreground objects closer together in the feature space could improve robustness against foreground variations. We plan to explore these approaches to further enhance the generalizability and applicability of our method to more complex, real-world scenarios.
>
> - \[7] James, Stephen, et al. "Sim-to-real via sim-to-sim: Data-efficient robotic grasping via randomized-to-canonical adaptation networks." Proceedings of the IEEE/CVF conference on computer vision and pattern recognition. 2019.

---

> > ### Comment · Reviewer_L53c · 2024-11-27
> >
> > Thanks to the authors for replying in the rebuttal. Meanwhile, this work still only handles background related distractions by design which is not generalized enough. Thereby the score is not changed.

---

### Official Review · Reviewer_qM91 · 2024-11-03

**Soundness:** 2
**Presentation:** 3
**Contribution:** 2
**Rating:** 5
**Confidence:** 5

**Summary:**

The paper introduces an approach to enhance MBRL in environments with visual distractions. Building upon the DREAMER framework, the authors propose a method called Segmentation Dreamer (SD). This method utilizes segmentation masks to focus on task-relevant components of the visual input, reducing the complexity of representation learning. To address the limitations of predicted masks, the authors introduce a selective L2 loss function.

**Strengths:**

To address the limitations of predicted masks, the authors propose a selective L2 loss function.
This is a thoughtful addition to improve the robustness of the model to segmentation errors.

**Weaknesses:**

1. While the paper presents an integration of segmentation masks into MBRL, it lacks a thorough comparison with existing works that also combine foundation models with RL. A more detailed analysis against these related works would strengthen the paper's claims and provide a clearer picture of the method's advantages and potential limitations[1, 2].
2. The generalizability of the approach across a wider range of real-world scenarios beyond the controlled environments should be considered. In real-world scenarios, distractions may closely resemble task-relevant features, potentially occluding critical information. The paper could benefit from evaluating the method in more realistic settings with complex distractions[3, 4].
3. The paper addresses some limitations, particularly regarding the dependence on accurate segmentation masks. However, a more detailed discussion on potential model biases, especially how segmentation errors might affect performance in untested environments, would be beneficial.



Reference:
[1] "Generalizable Visual Reinforcement Learning with Segment Anything Model." arXiv, 2024
[2] "Empowering Embodied Visual Tracking with Visual Foundation Models and Offline RL." European Conference on Computer Vision, 2024
[3] "Towards distraction-robust active visual tracking." International Conference on Machine Learning. PMLR, 2021.
[4] "Anti-distractor active object tracking in 3D environments." IEEE Transactions on Circuits and Systems for Video Technology 32.6 (2021): 3697-3707.

**Questions:**

1. Could the authors elaborate on how their approach of integrating segmentation masks into MBRL differs from and improves upon existing methods that combine foundation models with RL?
2. What are the authors' thoughts on the generalizability of their method to real-world tasks where distractions are inherently complex and closely intertwined with task-relevant features?
3. Can the authors discuss potential biases in their model that may arise from errors in segmentation masks? What strategies could mitigate the impact of segmentation errors in untested environments?
4. In environments where distractions and task-relevant features are dynamic, how does the method ensure consistent performance?
5. Is there a mechanism for the model to adapt to new environments or changing task dynamics without complete retraining?

---

> ### Author Response · Authors · 2024-11-25
>
> \[W1, Q1] Comparison with existing works that combine foundation models with RL
>
> - Thank you for suggesting these related works. Below, we summarize the key distinctions and advantages of our approach:
>
>   - **Use of Segmentation Models**: Existing works \[1, 2] utilize segmentation models for **preprocessing** input observations, meaning their success is heavily dependent on the quality of the segmentation model at test time. If the segmentation model fails during deployment (e.g., when encountering unfamiliar scenarios), it can disrupt the input and cause the agent to malfunction. In contrast, our method uses segmentation models for an auxiliary task, making the agent more robust to segmentation errors. The **instability** of using processed inputs is demonstrated in Table 1 of our paper, where “As Input” represents this family of prior approaches.
>
>   - **Training Overhead**: Because the quality of segmentation models is critical, methods like \[2, 5] require substantial effort to fine-tune segmentation models, often involving building simulations to generate synthetic datasets for fine-tuning. This introduces considerable time and resource overhead. In contrast, our approach can effectively use out-of-the-box segmentation models with minimal data (e.g., 1, 5, or 10 examples). Because we use segmentation masks for an auxiliary target, our method remains resilient to segmentation errors and eliminates the need for extensive fine-tuning during training.
>
>   - **Inference Efficiency**: Existing methods that rely on segmentation models for preprocessing add computational overhead during test time. This is undesirable for real-world scenarios where lightweight, efficient inference is critical, particularly in resource-constrained or on-device settings. Our method avoids this issue by limiting segmentation only to training, ensuring no additional overhead during deployment while retaining robustness.
>
> * While prior works have strengths such as simplifying input observations when the segmentation is accurate, our method uniquely balances robustness, simplicity, and efficiency. Additionally, we acknowledge that incorporating ideas from related works, such as instance-level segmentation and tracking (e.g., \[3, 4]), could enhance segmentation quality and make our method even more robust. Using canonical representation as an input can be especially robust to foreground variations which we can also use as another auxiliary task to be invariant to foreground perturbations. We plan to explore these directions in future work.
>
> * We have updated the paper to include additional prior work and provide a clearer picture of how our method compares. Please refer to Section 2 (marked in red).
>
> - \[5] John So, Amber Xie, Sunggoo Jung, Jeffrey Edlund, Rohan Thakker, Ali Agha-mohammadi, Pieter Abbeel, and Stephen James. Sim-to-real via sim-to-seg: End-to-end off-road autonomous driving without real data. In Conference on Robot Learning, 2022.
>
> \[W2, Q2] Generalizability to Real-World Tasks with Complex Distractions
>
> - Thank you for raising this insightful point. To demonstrate the promise of our method in more realistic scenarios, we evaluated our method against three additional types of distractions, including occlusions. Please refer to General Response 1 for further details.
>
> - While our method effectively handles a wide range of distractions, we acknowledge its limitations in scenarios where distractions closely resemble task-relevant features, as the segmentation model lacks knowledge of controllability. A promising direction for future work is to incorporate controllability knowledge into either the segmentation model or the RL agent, enabling them to differentiate between controllable and uncontrollable features. Another viable direction is to introduce a flag as an additional auxiliary task to explicitly identify controllable foreground objects, combined with video segmentation models capable of object tracking, as demonstrated in \[2] in the context of input preprocessing. These extensions represent tangible directions to improve our method’s generalizability and applicability to tasks with more intertwined distractions and task-relevant features.

---

> ### Author Response · Authors · 2024-11-25
>
> \[W3, Q3] Model bias
>
> - Thank you for raising this important point. Figure 9 in the Appendix (“SD^PerSAM\_1”) provides insight into how test-time segmentation errors might affect performance in untested environments. In this figure, the x-axis represents episodic segmentation errors at test-time, and the y-axis shows test returns. While our method does not rely on segmentation during testing, we performed segmentation ad hoc to assess its impact. Data points on the left (low test IoU) indicate high test-time segmentation errors from the model used during training. Despite these errors, our method exhibits stable performance, even in scenarios where the segmentation would have failed. This robustness highlights the method's resilience to unseen environments.
>
> - In this paper, we also propose two strategies to mitigate the impact of segmentation errors:
> - 1\) By only using segmentation predictions as training targets, we prevent failures that could be caused by deployment time prediction errors. This increase in robustness is made evident by comparing our SD^PerSAM\_1 method to the “As Input” variation in Table 1 and Figure 9. “As Input” applies segmentation mask predictions to deployment time observation inputs. Our proposed method avoids using segmentation masks in deployment and is thus less sensitive to segmentation accuracy.
>
> - 2\) We leverage Selective L2 loss to attempt to skip training on regions of reconstruction targets where the segmentation model is likely incorrect. The positive effect of reducing incorrect training examples on performance is shown in Table 1, where SD^PerSAM\_1 with Selective L2 loss shows a clear improvement over our Naive L2 loss ablation.
>
> - We hope that this answers your question concerning the impact of segmentation errors on test-time performance and how the effects of poor accuracy can be mitigated. Did we address your question fully?
>
> \[Q4] Handling Dynamic Distractions and Task-Relevant Features
>
> - **Dynamic Task-Relevant Features:** When task-relevant features, such as the position, shape, or appearance of foreground objects, shift during execution, our method can remain effective. If these changes are anticipated and incorporated as prior knowledge, segmentation models can be fine-tuned with larger datasets to account for such dynamics, ensuring consistent performance.
>
> - **Dynamic Distractions:** In cases where distractions, such as background or foreground elements, change unpredictably or frequently, our experiments show that our method is robust to significant background variations. Furthermore, our additional experimental results highlight the method’s potential in managing dynamic foreground elements effectively.
>
> - Looking forward, a promising direction for future work is to autonomously learn the distinction between distractions and task-relevant segmentation features directly from data, particularly in environments where these features change dynamically over time.
>
> \[Q5] Mechanism for Adapting to New Environments
>
> - Currently, our framework does not include a mechanism to adapt to new environments or changing task dynamics without complete retraining. This limitation aligns with the broader challenge in MBRL methods, which typically require fine-tuning on the target task or pre-training with multi-task learning to adapt effectively.
>
> - However, in our framework, if the source and target tasks share the same task-relevant objects, transfer can be facilitated. Developing a mechanism for seamless adaptation to new environments without retraining remains an intriguing direction for future work.

---

> > ### Comment · Reviewer_qM91 · 2024-11-28
> > **Decision Remains Unchanged**
> >
> > Thank you for your reply. However, after carefully considering your response, I regret to inform you that my view on the quality of this paper has not changed. I encourage you to continue to work on improving the manuscript. Good luck!

---

> > > ### Author Response · Authors · 2024-11-28
> > >
> > > Thank you for your thoughtful review and valuable feedback, which have greatly improved the quality of our work.
> > >
> > > To ensure that all updates in the revision are not missed, we would like to summarize the key additions:
> > >
> > > - **Section E in the Appendix** demonstrates the generalizability of our method to a variety of realistic distractions, including occlusions, color changes, and camera perturbations. These results highlight the robustness of our method, particularly in challenging scenarios involving occlusions.
> > >
> > > - **Figure 10 in Section D** investigates the impact of **segmentation bias** on downstream task performance at test time. This analysis provides deeper insights into how our method handles segmentation-related challenges:
> > >   - **Precision Plot (2nd Column):** Shows how methods address false positives.
> > >   - **Recall Plot (3rd Column):** Demonstrates how methods handle false negatives.
> > >   - **Variance of IoU (4th Column):** Reflects the temporal stability of segmentation models. Our method maintains high performance even when variance increases, showing its robustness to segmentation instability.
> > >   - **Gradient of IoU (5th Column):** Highlights flickering effects in segmentation models, calculated as IoU differences between consecutive frames. High gradient values signify increased flickering, and this analysis shows how our method effectively manages such challenges.
> > >
> > > - **Figure 10** demonstrates that our method outperforms models relying on preprocessed observations as input. SD exhibits resilience to both low recall (enabled by the selective L2 loss) and low precision in segmentation models, which are common challenges in real-world settings. Additionally, our method is robust to temporal instability, including flickering, offering strong performance even under these conditions.
> > >
> > > **Why this matters:**
> > > Your review insightfully highlighted the importance of robustness to segmentation model biases. Our additional experiments and analyses directly address this point, showing that SD consistently handles these challenges better than alternative methods. Furthermore, our findings provide actionable guidance for practitioners by identifying areas where segmentation models influence RL performance, such as through flickering or temporal instability.
> > >
> > > We will ensure that these discussions are incorporated into the final version to provide additional insights for readers.
> > >
> > > We kindly hope you will recognize the strength of our revisions and consider revisiting your rating of our work.
> > >
> > > If there are any remaining concerns or suggestions for improvement, we would greatly appreciate your feedback and will gladly address them.
> > >
> > > Kind regards,

---

### Official Review · Reviewer_yfcf · 2024-11-03

**Soundness:** 3
**Presentation:** 3
**Contribution:** 2
**Rating:** 6
**Confidence:** 3

**Summary:**

This paper proposes Segmentation Dreamer (SD), an approach to learn task relevant features in model-based RL framework by using segmentation mask from prior domain knowledge as an auxiliary objective. The proposed approach only reconstruct task relevant components instead of entire original image to avoid noisy learning signals from distractions. Additionally a selective l2 loss is proposed to handle error predictions from segmentation masks. Experiments are performed on visual control benchmarks including DMC and Meta-World with visual distractions by replacing them with diverse background scenes. Experiments demonstrate that SD outperforms Dreamer and other baselines in better sample efficiency and convergence performance. Ablations show effectiveness of each component, specifically focusing on segmentation quality and usefulness of proposed selective l2 loss.

**Strengths:**

1. Paper is well-written, offering clarity and easy to understand
2. The motivation of only reconstructing task-relevant parts in image is reasonable. The proposed approach can handle potential error of adapting pre-trained segmentation model to visual RL domains
3. Extensive experiments including baseline comparisons and ablation analysis demonstrate benefits of proposed approach in improving sample-efficiency

**Weaknesses:**

1. This framework requires significant prior knowledge about which part is task-relevant in the image. This may limit the scope of potential usage of the proposed approach, where domain knowledge may be ambiguous or hard to obtain.
2. Since only parts of image is reconstructed specified by mask, this problem setting is simpler than the original, which may be hard to directly compare to other baselines
3. Experiments are limited to performance of visual control tasks; we do not know how good pre-trained segmentation models are adapted to downstream domains.

**Questions:**

1. Since segmentation quality is important to task performance (Figure 3b), what will happen if more samples, than 1 or 5-10, are used in fine-tuning pre-trained segmentation models to downstream domains? Will the proposed selective l2 loss become less important if segmentation model improves with more training samples?
2. How is binary mask predictor head trained, i.e. what is loss function objective here? What will happen if binary mask decoder is trained but selective l2 loss is not used in learning world model, i.e. is the mask prediction task or addressing segmentation model errors more important in this framework?
3. In terms of $mask_\mathbf{SD}$ and $mask_\mathbf{FM}$, could authors visualize how these segmentations look like in order to understand what are similarity and differences in predictions from adapting pre-trained models and those from the process of learning world models?

---

> ### Author Response · Authors · 2024-11-25
>
> \[W1] Prior Knowledge Requirement and Its Potential Limitations
>
> - Yes. Our method indeed assumes that task-relevant parts of image observations can be identified using prior knowledge. This aligns with how humans leverage prior knowledge, such as affordances \[6], to efficiently solve tasks. We believe there are many domains where obtaining prior knowledge is straightforward, such as robotics tasks involving object manipulation or navigation, where task-relevant features are naturally evident. In such cases, our method suggests an effective and efficient interface to incorporate this prior knowledge, significantly simplifying the learning process while maintaining minimal computational overhead.
>
> - That said, we recognize that in domains where prior knowledge is ambiguous or difficult to obtain, this could limit the applicability of our approach. A promising direction for future work is to automatically learn task-relevant knowledge from data during training, using segmentation maps as an intermediate representation. This would expand the applicability of our method to more complex environments where task-relevant components are not explicitly known.
>
> - \[6] Shikhar Bahl, Russell Mendonca, Lili Chen, Unnat Jain, Deepak Pathak. Affordances from Human Videos as a Versatile Representation for Robotics. In  Computer Vision and Pattern Recognition, 2023.
>
> \[W2] Simpler problem setup?
>
> - By only reconstructing task-relevant portions of images we are indeed simplifying the functionality that the world model has to learn. This is an intended benefit of our proposed method that we see as a strength. Our comparisons with baselines like standard Dreamer show the benefit of leveraging prior knowledge to create a purposefully more tractable learning task via task-relevant reconstruction. By leveraging task-relevant reconstruction to only model the necessary components of observations, we can train a world model and agent more robust to task-irrelevant visual perturbations.
>
> - Moreover, prior works \[1, 2] have also leveraged segmentation models with task-specific prior knowledge to inform segmentation. However, these methods use segmentation for preprocessing input observations, which has been shown to be unstable at test time. This instability is evident in Table 1, where “As Input” represents this family of methods, demonstrating their vulnerability in untested environments. Our careful design eliminates these limitations, ensuring robustness and efficiency while solving the same underlying problem.
>
> \[W3] Limited to visual control task
>
> - While our method leverages segmentation models specifically for visual control tasks, these tasks encompass a wide range of applications, including object manipulation and navigation. Visual observations are a commonly used multimodal representation for perceiving the world, making our approach valuable for shaping features to focus on task-relevant image components in such scenarios.
>
> - To evaluate the performance of adapted segmentation models in downstream domains, we suggest two approaches:
>
>   - Qualitative Evaluation: By inferring the fine-tuned segmentation model on test scenarios, one can visually inspect segmentation quality. This provides insights into how well the RL agent might perform, given the observed correlation between segmentation quality and RL test-time performance (Fig. 3b).
>
>   - Quantitative Evaluation: Creating an evaluation set to quantify segmentation model performance offers a practical way to analyze its quality and assess its adaptation to downstream tasks as part of ad-hoc analysis, as illustrated in Fig. 9 of the Appendix.

---

> ### Author Response · Authors · 2024-11-25
>
> \[Q1] The effect of the proposed l2 loss if the segmentation model improves
>
> - As the segmentation model improves with more training samples, the performance gap between the naive L2 loss and the selective L2 loss is expected to decrease. However, this improvement comes at the cost of additional computational overhead during training, including the need for larger datasets and extended fine-tuning of the segmentation model.
>
> - Furthermore, our additional experiments added during this discussion phase show that the selective L2 loss remains effective in scenarios involving occlusions. In these cases, SD with the selective L2 loss can recover occluded parts of foreground objects and maintain robustness. This underscores the versatility and value of selective L2 loss across various challenging conditions. We plan to include these findings in the revision to provide additional evidence of its effectiveness in different scenarios.
>
> \[Q2] The effect of mask prediction
>
> - The binary mask predictor is trained using the standard L2 loss.
>
> - When both the RGB and binary mask decoders are trained with the standard L2 loss, the performance significantly drops compared to our method. This is partially because there is no mechanism to correct errors in the RGB decoder, allowing incorrect signals to propagate throughout training. Additionally, binary mask prediction does not serve as an effective auxiliary task, as it tends to generate overly abstract and coarse features that fail to support downstream tasks properly.
>
> - These findings highlights two important implications:
>
>   - **Addressing Segmentation Errors**: Correcting segmentation model errors is critical for achieving robust performance. This highlights the value of the selective L2 loss, which effectively mitigates segmentation inaccuracies and prevents error propagation during training.
>
>   - **Careful Design of Auxiliary Losses**: The auxiliary loss must be thoughtfully designed to balance feature abstraction and task relevance. Selective L2 loss provides this balance by selectively focusing on regions where predictions are more reliable, enhancing overall learning stability.
>
> - We will include these insights in the revision to provide a deeper understanding of the importance of addressing segmentation errors and designing auxiliary tasks aligned with the needs of our framework.
>
> |                         |                              |                                       |                                  |
> | ----------------------- | ---------------------------- | ------------------------------------- | -------------------------------- |
> |  | **RGB** **w/ naive L2 loss** | **Binary & RGB** **w/ naive L2 loss** | **RGB** **w/ selective L2 loss** |
> | Walker Run              | 557 ± 51                     | 327 ± 65                              | 730 ± 13                         |
> | Cartpole Swing          | 719 ± 62                     | 379 ± 35                              | 730 ± 75                         |

---

> > ### Comment · Reviewer_yfcf · 2024-11-27
> >
> > I would like to thank authors for the rebuttal. Ablations about selective L2 loss and additional experiments on different types of visual distractions are helpful to show effectiveness and some generalization capability of this approach. Authors are suggested to include additional experiments in later versions to provide further insights.
> >
> > Overall I think this manuscript proposes an approach to utilize pre-trained segmentation model to address visual distractions in control tasks and demonstrate its effectiveness in empirical evaluations, though technical contribution is somewhat limited and assumptions of task-specific prior knowledge may limit its applicability to more complex environment. Changed to weak accept.

---

### Official Review · Reviewer_RWSA · 2024-11-04

**Soundness:** 3
**Presentation:** 3
**Contribution:** 3
**Rating:** 6
**Confidence:** 4

**Summary:**

This paper addresses the challenge of training Model-Based Reinforcement Learning (MBRL) agents in visually distracting environments, where standard approaches struggle with representation learning due to high visual variability. Building on the DREAMER framework, the authors propose Segmentation Dreamer (SD), an auxiliary task that uses segmentation masks to focus representation learning on task-relevant elements, thereby simplifying latent encoding by excluding irrelevant objects. In experiments with visually complex tasks, SD significantly improves sample efficiency, performance, and robustness, particularly in sparse reward scenarios, enabling effective agent training without extensive reward engineering.

**Strengths:**

* This paper is well-written and easy to follow. The idea of using the segmentation to capture the task-relavent information is reasonable.

* The literature review is sufficient and the authors have adequately discussed the major difference between the proposed method and existing works.

* The evaluations of the proposed method are comprehensive. The authors compare the proposed methods with several state-of-the-art algorithms and evaluations are conducted on two benchmarks.

**Weaknesses:**

* Although the authors conduct experiments on several benchmarks, I still have concerns about the generalization ability. In DMControl tasks, the agent’s visual state consistently appears at the center of the image, and the fixed camera makes segmenting the agent's body relatively straightforward. Could the authors provide additional visualized results for Meta-World and other more complex tasks? For instance, how would segmentation handle the robotic arm and objects in these more challenging scenarios?

* The segmentation process will inevitably introduce additional computational overhead during the model training. For example, the proposed method need masks of the agent body, which is obtained from a fine-tuned SAM model.

**Questions:**

1. How does the quality of the mask affect the results of the proposed method? The authors propose a selective L2 loss during training. Can the authors provide results about using native L2 loss and selective L2 loss for training respectively?

---

> ### Author Response · Authors · 2024-11-25
>
> \[W1] Segmentation Quality in More Challenging Scenarios
>
> - To address concerns about generalization, we conducted additional experiments on DMC with three different distraction types, including camera view changes. Both the segmentation models and the SD RL algorithm demonstrated robustness to these perturbations. Please refer to General Response 1 for details.
>
> - Additionally, Figures 16–18 in the Appendix show segmentation results on more complex tasks in Meta-World, involving multiple objects and occlusions. The segmentation models were fine-tuned using only 10 data points, yet they performed robustly under challenging conditions such as unseen poses and occlusions (e.g., robotic arm occluding itself or objects). This robustness can be attributed to the models’ pre-training on large-scale datasets. Naturally, some segmentation errors still occur, but these are effectively managed in our framework using selective L2 loss.
>
> \[W2] Training time computational overhead
>
> - Yes. Incorporating segmentation models during training introduces additional computational overhead. To quantify this, we conducted an analysis on DMC-1M using an NVIDIA RTX A4500 and SegFormer with the MiT-b0 backbone, as used in our main experiments:
>
>   - **Training Time**: Standard Dreamer required 539 minutes (≈9 hours), while our method took 813 minutes (≈13.5 hours), reflecting a \~50% increase. These numbers are averaged over 8 runs. One important point to mention is that our current implementation is relatively unoptimized, where images from experience collection are each sequentially processed individually by the segmentation model. With further code optimizations, such as batch inference and asynchronous image processing, we expect a significant acceleration in training time. For example, using batch inference with a batch size of four (matching the number of simulators running), the additional training time would reduce to approximately 68.5 minutes, corresponding to only a \~12% increase instead of the current \~50%.
>
>   - **VRAM Usage**: Standard Dreamer utilized 3200MiB VRAM, while our method required an additional 1113MiB, a \~35% increase.
>
> - We believe this computational overhead during training is a reasonable trade-off for achieving a significantly better policy. Importantly, our method introduces no additional overhead during test time, unlike approaches that incorporate segmentation modules for preprocessing in the inference pipeline. Test-time efficiency is particularly critical in scenarios when delays are unacceptable or on-device inference is required, where additional computational demands are highly undesirable. By using segmentation masks for an auxiliary target during training rather than for preprocessing inputs, our approach avoids these issues while maintaining effectiveness.
>
> \[Q1] How does the quality of the mask affect the results of the proposed method?
>
> - Figure 3b plots the training-time segmentation quality against the RL agent’s test-time performance. We observe that better segmentation tends to lead to higher RL performance, as accurate targets better highlight task-relevant components. This suggests that improved segmentation models can enhance agent performance without ground-truth masks.
>
> \[Q2] Results about using native L2 loss and selective L2 loss for training respectively?
>
> - Table 1 demonstrates that using the selective L2 loss (SD^PerSAM\_1) consistently outperforms the naive L2 variants, particularly in complex tasks such as Cheetah Run and Walker Run. Segmentation models often miss embodiment components. With naive L2 loss, the model replicates these errors, leading to incomplete latent representations and harming dynamics learning. In contrast, the selective L2 loss addresses this issue by skipping L2 computation in regions where PerSAM targets are likely incorrect. Additionally, our experiments on foreground distractions further reinforce the effectiveness of the selective L2 loss in enhancing robustness when occlusions affect the foreground agent.

---

> > ### Comment · Reviewer_RWSA · 2024-11-27
> >
> > I thank the authors for the rebuttal. Overall, this is an interesting paper with good empirical results, but the incorporated segmentation model indeed makes the whole pipeline looks straightforward and introduces more computational cost. As a result, I decide to keep my score as borderline accept and will not object to acceptance of this paper.

---

> > > ### Author Response · Authors · 2024-11-27
> > >
> > > Thank you for recognizing the strengths of our work and providing constructive feedback to help us refine it. If there are any additional questions or clarifications, please let us know, and we will be happy to address them during the discussion period.

---

### Official Review · Reviewer_3S6K · 2024-11-08

**Soundness:** 2
**Presentation:** 3
**Contribution:** 2
**Rating:** 5
**Confidence:** 4

**Summary:**

This paper proposes Segmentation Dreamer, a model-based reinforcement learning method that extends the Dreamer framework by introducing an auxiliary task focused on predicting task-related segmentation masks. Specifically, the reconstruction task in Dreamer is modified to work with images that contain task-relevant parts, rather than reconstructing the entire image. The authors evaluate the method on two RL benchmarks: DeepMind Control and Meta-World, demonstrating its effectiveness in these environments.

**Strengths:**

1. Clear Presentation: The paper is well-written, with clear and easy-to-understand figures, text, and mathematical formulations. The explanation of the model and its components is thorough, making it accessible to both newcomers and experienced researchers in the field.

2. Intuitive Idea: The core idea of the paper—introducing task-relevant segmentation as an auxiliary task—is intuitively appealing. It draws inspiration from human perception, where individuals tend to focus on the most relevant parts of their environment when performing a specific task. This concept feels natural and aligns with cognitive science.

3. Solid Experimental Comparison: The authors conduct experiments for comparing with some strong baselines, and provide detailed ablations for SD's designs.

**Weaknesses:**

1. Limited Scope of Contribution: One of the main weaknesses of the paper is that the contribution feels somewhat narrow. While the introduction of task-relevant segmentation masks is an interesting idea, it could potentially have a broader application beyond Dreamer or visual distraction tasks. Other model-based RL frameworks, such as transformer-based models (e.g., Trajectory Transformer), also use reconstruction as an auxiliary task, meaning the innovation may not be entirely novel. The applicability of this approach to other RL models and tasks remains underexplored, which limits its perceived impact.

2. Stability Concerns in Meta-World: The experimental results on Meta-World show large reward variance, which raises concerns about the stability of the model. This suggests that the proposed approach may struggle with robustness, particularly in more complex or dynamic environments. A deeper analysis of the reasons for this instability and potential solutions would strengthen the paper.

3. Limited Scope of Experiments: The experiments are conducted on relatively simple and controlled environments (DMC and Meta-World), which might not fully capture the complexity and variability of real-world tasks. The lack of experiments in more complex or high-dimensional settings limits the generalizability of the findings. It would be helpful to see how the method performs on more challenging tasks or in real-world applications where the environment is less predictable.

**Questions:**

How to ensure that the segmentation masks correctly identify task-relevant parts of the image, especially when the foundation model (used for segmentation) does not inherently understand what is important for the task. In some cases, the background details may be crucial for the task, and it is not clear how the model can distinguish between what is task-relevant and what is not. How can you ensure that the masking process does not remove essential information, especially in environments where the task might require nuanced understanding of the background or context? In other words, how can the sparse rewards guide good segmentation masks prediction, when they are not enough at guiding good policies?

The variance in the rewards observed in the Meta-World experiments could be indicative of instability in the model. Can the authors provide more insights into why this happens and how this issue might be mitigated? For example, are there any hyperparameters or training strategies that could be tuned to improve stability across environments?

---

> ### Author Response · Authors · 2024-11-25
>
> \[W1] Broader application beyond Dreamer or visual distraction tasks
>
> - The focus of our paper is on addressing distractions in visual control tasks, a highly challenging problem that reflects the complexities of real-world scenarios. Numerous prior works (e.g., Deng et al., 2022; Nguyen et al., 2021; Zhang et al., 2021) have also concentrated exclusively on this specific issue.
>
> - That said, we agree that demonstrating the versatility of our method would make it even more impactful.
>
>   - **Applicability to Other MBRL Frameworks**: Indeed, our auxiliary task is not limited to Dreamer and could potentially be integrated into other reconstruction-based model-based RL (MBRL) frameworks, including transformer-based models such as Trajectory Transformers (Janner et al., 2021) or STORM (Zhang et al., 2023). While these frameworks rely on reconstruction as an auxiliary task, they likely struggle in distracting environments because naive reconstruction preserves unnecessary information, hindering agent learning. Our auxiliary task can address this issue by enabling those methods to focus on agent-centric features and mitigate distractions through feature shaping with segmentation models. Since our approach is orthogonal to advancements in MBRL architectures, it serves as a complementary enhancement. We see great potential for future work to apply our method to other architectures.
>
>   - **Use Beyond Visual Distraction Tasks**: The idea of using segmented images as auxiliary targets extends beyond addressing visual distractions. For instance, it can facilitate object-centric representation learning by reconstructing the segmentation of individual objects. This concept has been explored by FOCUS (Ferraro et al., 2023), which demonstrated its effectiveness using ground-truth masks. We believe that by leveraging segmentation models, our approach can support disentangled representation learning, even when ground-truth masks are unavailable. Additionally, it has the potential to address occlusions often encountered in object manipulation tasks through the use of selective L2 loss.
>
> - In summary, while this work specifically focuses on visual distractions, its underlying concepts are versatile and applicable to a wider range of tasks and frameworks, offering promising directions for future research.
>
> \[W2, Q2] Reward variance in Meta-world
>
> - Thanks, this is a good point that we will elaborate on better in subsequent revisions. In figure 5, while “SD^SegFormer\_10” exhibits higher performance than other baselines, it also shows slightly more variance. The primary contributing factor to this variance is the accuracy of the segmentation model we pair with Segmentation Dreamer. We can improve the variance by A) using a better segmentation foundation model, B) using more finetuning image/mask examples (we used 10 in this case), and C) using a more principled approach to select a diverse set of representative finetuning image/mask pairs. In our current experiments, we selected finetuning data points by rolling out random behavior in each environment and selecting random frames from the trajectories. This selection method already effectively showcases the benefit of Segmentation Dreamer over other approaches; however, refining the data selection process could further enhance performance stability which we leave for future work.
>
> - Importantly, a demonstration of Segmentation Dreamer performance and variance with a better (perfect) segmentation mask model is shown “SD^GT”.

---

> ### Author Response · Authors · 2024-11-25
>
> \[W3] Evaluation on more challenging tasks or in real-world applications
>
> - While DMC is a simulated and controlled environment, solving tasks with distractions in these settings remains a challenging problem. Many prior works on DMC struggle significantly, especially when rewards are sparse, whereas our method demonstrates strong performance under such conditions.
>
> - Beyond DMC, we evaluate our method on Meta-World, a widely recognized and challenging benchmark for RL. Meta-World involves tasks requiring object manipulation with multiple objects, frequent occlusions, and varying object poses, all of which introduce significant complexity. Meta-World resembles real-world scenarios, where agents must handle dynamic interactions and multi-step processes to achieve their goals.
>
> - To further evaluate our method’s robustness, we also present preliminary experiments on three additional types of distractions to increase environmental complexity. Please refer to General Response 1 for details. Expanding to real-world applications, where unpredictability and variability are even greater, is a promising direction that we plan to explore in future work.
>
> \[Q1] Ensuring Segmentation Masks Identify Task-Relevant Parts
>
> - In our current framework, the foundation model does not inherently understand task relevance. Instead, it is informed about the task using a few labeled examples provided before the RL training loop begins. Our paper focuses on designing the most effective way to integrate MBRL with segmentation models, given examples of a few correct task-relevance masks as prior knowledge.
>
> - By reconstructing agent-centric parts of the observation with the help of the segmentation model, our method successfully solves tasks even in scenarios with sparse rewards, where other baselines fail completely (see Fig. 3a, Cartpole Swing Sparse).
>
> - Making the segmentation model inherently aware of task relevance and enabling it to learn and adjust task-relevant features directly from rewards during RL training remains an exciting direction for future research.

---

> ### Comment · Area_Chair_gfes · 2024-11-30
>
> Dear reviewer 3S6K,
>
> Thanks for your efforts in reviewing this paper. Could you please let us know if the authors have addressed your comments? Or if you still have any other concerns?
>
> Best regards,
> Your AC

---

> ### Comment · Reviewer_3S6K · 2024-12-01
>
> Thanks for the responses. I see the authors provide some justifications for the scope of this paper, but this does not convince me. The paper's contribution is still limited when only focusing on Dreamer and current simulated distraction environments. Besides, simply introducing pre-trained segmentation models without letting the model be aware of task information also makes the method less insightful and too straightforward. This does not address the fundamental problem and challenges in visual control tasks with distractions. Therefore, I decide to keep my score.

---

### Author Response · Authors · 2024-11-25
**General Response**

Dear Reviewers,

We sincerely thank you for your valuable feedback and insightful comments. We are particularly grateful for recognizing our work’s contribution to addressing background distractions in visual control tasks, a well-known and challenging problem.

- Reviewer 3S6K’s positive feedback on the core idea of introducing task-relevant segmentation as an auxiliary task is especially encouraging.

- We appreciate Reviewer RWSA and Reviewer yfcf’s acknowledgment of our efforts in providing a comprehensive evaluation of our method and baselines.

- We are also grateful for Reviewer qM91 and Reviewer L53c’s positive comments on the selective L2 loss function and its role in increasing robustness.

We hope this discussion addresses all your concerns and further strengthens the paper. If the reviewers find our responses satisfactory, we kindly request that they consider increasing their scores by the end of the rebuttal period.


## 1. Generalization to more complex, realistic environments

While our current experiments primarily focus on background distractions, the proposed method has the potential to be extended to handle other types of distractions, including foreground occlusions, color variations, and changes in camera angle.

To better reflect real-world scenarios, we have added evaluations for our method on three additional types of visual control distractions:

1. **Foreground distractions**: Potentially occlude task-relevant objects.

2. **Color variations in foreground objects**: Simulating changes due to environmental factors like lighting.

3. **Camera angle changes**: Altering the perspective of the scene.

For sample images illustrating perturbations and detailed experimental setups, please refer to **Appendix Figures 11–14**. To handle a variety of distractions at test time, both the segmentation models and SD are trained and evaluated in domain-randomized environments.

The success of our method hinges on two key aspects, which we analyze below:

1. **Robustness of Visual Foundation Models (Segmentation Models)**: Can segmentation models remain robust to these perturbations and distractions?

- Answer: Yes, our results demonstrate that segmentation models are highly effective at identifying objects of interest, even in the presence of these distractions. This aligns with observations from our primary experiments on background distractions. Leveraging segmentation models remains a robust strategy for training RL agents under these conditions. For segmentation quality, please refer to **Figures 11–14 in the Appendix**. Additionally, we plan to include quantitative evaluations of segmentation model performance in the revision to provide further insights.

2. **Capacity of SD to Handle Perturbations**: Can SD manage perturbations that segmentation models cannot filter?

- Answer: Unlike background distractions, perturbations in color and camera angle cannot be fully filtered out and are retained in the target. While this might raise concerns about wasted capacity due to spurious information, our results (see Table below) show that SD effectively handles these perturbations. The table summarizes the test returns (mean and standard error) over four runs, where 100 data points were used for fine-tuning. Notably, our method, by focusing on agent-centric features, outperforms the standard Dreamer trained in the standard environment. Additionally, the selective L2 loss proves highly effective in handling occlusions, enabling the recovery of occluded foreground agents. This underscores the versatility and value of the selective L2 loss across multiple scenarios. While we designed SD primarily to mitigate visual distractions outside of task-relevant objects, these new results show that it is also robust to other realistic forms of distractions.

|                |                                        |                             |                                                     |           |                  |                             |
| -------------- | -------------------------------------- | --------------------------- | --------------------------------------------------- | --------- | ---------------- | --------------------------- |
|                | Standard environment (Standard Dreamer) | **Foreground distractions** | **Foreground distractions** (**No** selective loss) | **Color** | **Camera angle** | **Background with GT mask** |
| Walker         | 752 ± 9                                | 740 ± 4                     | 688 ± 37                                            | 761 ± 3   | 752 ± 10         | 719 ± 19                    |
| Cartpole Swing | 818 ± 52                               | 860 ± 2                     | 852 ± 7                                             | 870 ± 4   | 863 ± 4          | 869 ± 4                     |

---

### Meta-Review · Area_Chair_gfes · 2024-12-21

**Metareview:**

This paper introduces Segmentation Dreamer (SD), an extension of the DREAMER framework designed to improve Model-Based Reinforcement Learning (MBRL) in visually distracting environments. SD focuses representation learning on task-relevant components of the visual input by using segmentation masks, simplifying latent encoding and avoiding noisy signals from irrelevant objects. The method is evaluated on tasks from the DeepMind Control and Meta-World benchmarks, demonstrating improved sample efficiency and performance.

The strengths of this paper include a clear presentation with easy-to-understand explanations and visualizations, as well as an intuitive idea of using segmentation masks to focus on task-relevant components. The main weaknesses include limited generalizability, as the experiments are primarily conducted in controlled environments, the method's dependence on prior knowledge to identify task-relevant components, and its computational cost.

This paper is decided to be rejected, with an average score of 5.4, including two borderline positive scores and three borderline negative scores.

**Additional Comments On Reviewer Discussion:**

Reviewer yfcf changed their score from 5 to 6 due to additional experiments and the method's effectiveness in empirical evaluations, though further experiments would still be welcomed. No other reviewers changed their scores.

No reviewer explicitly championed the paper during the post-rebuttal AC-reviewer discussion.

---

### Decision · Program_Chairs · 2025-01-22

Reject